# Dykstra's Algorithm, ADMM, and Coordinate Descent: Connections, Insights, and Extensions

**Ryan J. Tibshirani**
Department of Statistics and Machine Learning Department
Carnegie Mellon University
Pittsburgh, PA 15213
ryantibs@stat.cmu.edu

## Abstract

We study connections between Dykstra's algorithm for projecting onto an intersection of convex sets, the augmented Lagrangian method of multipliers or ADMM, and block coordinate descent. We prove that coordinate descent for a regularized regression problem, in which the penalty is a separable sum of support functions, is exactly equivalent to Dykstra's algorithm applied to the dual problem. ADMM on the dual problem is also seen to be equivalent, in the special case of two sets, with one being a linear subspace. These connections, aside from being interesting in their own right, suggest new ways of analyzing and extending coordinate descent. For example, from existing convergence theory on Dykstra's algorithm over polyhedra, we discern that coordinate descent for the lasso problem converges at an (asymptotically) linear rate. We also develop two parallel versions of coordinate descent, based on the Dykstra and ADMM connections.

## 1 Introduction

In this paper, we study two seemingly unrelated but closely connected convex optimization problems, and associated algorithms. The first is the *best approximation problem*: given closed, convex sets $C_1, \ldots, C_d \subseteq \mathbb{R}^n$ and $y \in \mathbb{R}^n$, we seek the point in $C_1 \cap \cdots \cap C_d$ (assumed nonempty) closest to $y$, and solve

$$\min_{u \in \mathbb{R}^n} \ \|y - u\|_2^2 \quad \text{subject to} \quad u \in C_1 \cap \cdots \cap C_d. \tag{1}$$

The second problem is the *regularized regression problem*: given a response $y \in \mathbb{R}^n$ and predictors $X \in \mathbb{R}^{n \times p}$, and a block decomposition $X_i \in \mathbb{R}^{n \times p_i}$, $i = 1, \ldots, d$ of the columns of $X$ (i.e., these could be columns, or groups of columns), we build a working linear model by applying blockwise regularization over the coefficients, and solve

$$\min_{w \in \mathbb{R}^p} \ \frac{1}{2}\|y - Xw\|_2^2 + \sum_{i=1}^{d} h_i(w_i), \tag{2}$$

where $h_i : \mathbb{R}^{p_i} \to \mathbb{R}$, $i = 1, \ldots, d$ are convex functions, and we write $w_i \in \mathbb{R}^{p_i}$, $i = 1, \ldots, d$ for the appropriate block decomposition of a coefficient vector $w \in \mathbb{R}^p$ (so that $Xw = \sum_{i=1}^{d} X_i w_i$).

Two well-studied algorithms for problems (1), (2) are *Dykstra's algorithm* (Dykstra, 1983; Boyle and Dykstra, 1986) and *(block) coordinate descent* (Warga, 1963; Bertsekas and Tsitsiklis, 1989; Tseng, 1990), respectively. The jumping-off point for our work in this paper is the following fact: *these two algorithms are equivalent for solving* (1) *and* (2). That is, for a particular relationship between the sets $C_1, \ldots, C_d$ and penalty functions $h_1, \ldots, h_d$, the problems (1) and (2) are duals of each other, and Dykstra's algorithm on the primal problem (1) is exactly the same as coordinate descent on the dual problem (2). We provide details in Section 2.

This equivalence between Dykstra's algorithm and coordinate descent can be essentially found in the optimization literature, dating back to the late 1980s, and possibly earlier. (We say "essentially" here because, to our knowledge, this equivalence has not been stated for a general regression matrix $X$, and only in the special case $X = I$; but, in truth, the extension to a general matrix $X$ is fairly straightforward.) Though this equivalence has been cited and discussed in various ways over the years, we feel that it is not as well-known as it should be, especially in light of the recent resurgence of interest in coordinate descent methods. We revisit the connection between Dykstra's algorithm and coordinate descent, and draw further connections to a third method—the *augmented Lagrangian method of multipliers* or ADMM (Glowinski and Marroco, 1975; Gabay and Mercier, 1976)—that has also received a great deal of attention recently. While these basic connections are interesting in their own right, they also have important implications for analyzing and extending coordinate descent. Below we give a summary of our contributions.

1. We prove in Section 2 (under a particular relationship between $C_1, \ldots, C_d$ and $h_1, \ldots, h_d$) that Dykstra's algorithm for (1) is equivalent to block coordinate descent for (2). (This is a mild generalization of the previously known connection when $X = I$.)

2. We also show in Section 2 that ADMM is closely connected to Dykstra's algorithm, in that ADMM for (1), when $d = 2$ and $C_1$ is a linear subspace, matches Dykstra's algorithm.

3. Leveraging existing results on the convergence of Dykstra's algorithm for an intersection of halfspaces, we establish in Section 3 that coordinate descent for the lasso problem has an (asymptotically) linear rate of convergence, regardless of the dimensions of $X$ (i.e., without assumptions about strong convexity of the problem). We derive two different explicit forms for the error constant, which shed light onto how correlations among the predictor variables affect the speed of convergence.

4. Appealing to parallel versions of Dykstra's algorithm and ADMM, we present in Section 4 two parallel versions of coordinate descent (each guaranteed to converge in full generality).

5. We extend in Section 5 the equivalence between coordinate descent and Dykstra's algorithm to the case of nonquadratic loss in (2), i.e., non-Euclidean projection in (1). This leads to a Dykstra-based parallel version of coordinate descent for (separably regularized) problems with nonquadratic loss, and we also derive an alternative ADMM-based parallel version of coordinate descent for the same class of problems.

## 2 Preliminaries and connections

**Dykstra's algorithm.** Dykstra's algorithm was first proposed by Dykstra (1983), and was extended to Hilbert spaces by Boyle and Dykstra (1986). Since these seminal papers, a number of works have analyzed and extended Dykstra's algorithm in various interesting ways. We will reference many of these works in the coming sections, when we discuss connections between Dykstra's algorithm and other methods; for other developments, see the comprehensive books Deutsch (2001); Bauschke and Combettes (2011) and review article Bauschke and Koch (2013).

Dykstra's algorithm for the best approximation problem (1) can be described as follows. We initialize $u^{(0)} = y$, $z^{(-d+1)} = \cdots = z^{(0)} = 0$, and then repeat, for $k = 1, 2, 3, \ldots$:

$$
\begin{aligned}
u^{(k)} &= P_{C_{[k]}}(u^{(k-1)} + z^{(k-d)}), \\
z^{(k)} &= u^{(k-1)} + z^{(k-d)} - u^{(k)},
\end{aligned}
\tag{3}
$$

where $P_C(x) = \operatorname{argmin}_{c \in C} \|x - c\|_2^2$ denotes the (Euclidean) projection of $x$ onto a closed, convex set $C$, and $[\cdot]$ denotes the modulo operator taking values in $\{1, \ldots, d\}$. What differentiates Dykstra's algorithm from the classical *alternating projections method* of von Neumann (1950); Halperin (1962) is the sequence of (what we may call) dual variables $z^{(k)}$, $k = 1, 2, 3, \ldots$. These track, in a cyclic fashion, the residuals from projecting onto $C_1, \ldots, C_d$. The simpler alternating projections method will always converge to a feasible point in $C_1 \cap \cdots \cap C_d$, but will not necessarily converge to the solution in (1) unless $C_1, \ldots, C_d$ are subspaces (in which case alternating projections and Dykstra's algorithm coincide). Meanwhile, Dykstra's algorithm converges in general (for any closed, convex sets $C_1, \ldots, C_d$ with nonempty intersection, see, e.g., Boyle and Dykstra (1986); Han (1988); Gaffke and Mathar (1989)). We note that Dykstra's algorithm (3) can be rewritten in a different form, which

will be helpful for future comparisons. First, we initialize $u_d^{(0)} = y$, $z_1^{(0)} = \cdots = z_d^{(0)} = 0$, and then repeat, for $k = 1, 2, 3, \ldots$:

$$
\left.
\begin{aligned}
u_0^{(k)} &= u_d^{(k-1)}, \\
u_i^{(k)} &= P_{C_i}(u_{i-1}^{(k)} + z_i^{(k-1)}), \\
z_i^{(k)} &= u_{i-1}^{(k)} + z_i^{(k-1)} - u_i^{(k)},
\end{aligned}
\right\} \quad \text{for } i = 1, \ldots, d.
\tag{4}
$$

**Coordinate descent.** Coordinate descent methods have a long history in optimization, and have been studied and discussed in early papers and books such as Warga (1963); Ortega and Rheinboldt (1970); Luenberger (1973); Auslender (1976); Bertsekas and Tsitsiklis (1989), though coordinate descent was still likely in use much earlier. (Of course, for solving linear systems, coordinate descent reduces to Gauss-Seidel iterations, which dates back to the 1800s.) Some key papers analyzing the convergence of coordinate descent methods are Tseng and Bertsekas (1987); Tseng (1990); Luo and Tseng (1992, 1993); Tseng (2001). In the last 10 or 15 years, a considerable interest in coordinate descent has developed across the optimization community. With the flurry of recent work, it would be difficult to give a thorough account of the recent progress on the topic. To give just a few examples, recent developments include finite-time (nonasymptotic) convergence rates for coordinate descent, and exciting extensions such as accelerated, parallel, and distributed versions of coordinate descent. We refer to Wright (2015), an excellent survey that describes this recent progress.

In (block) coordinate descent[1] for (2), we initialize say $w^{(0)} = 0$, and repeat, for $k = 1, 2, 3, \ldots$:

$$
w_i^{(k)} = \operatorname*{argmin}_{w_i \in \mathbb{R}^{p_i}} \frac{1}{2} \left\| y - \sum_{j<i} X_j w_j^{(k)} - \sum_{j>i} X_j w_j^{(k-1)} - X_i w_i \right\|_2^2 + h_i(w_i), \quad i = 1, \ldots, d. \tag{5}
$$

We assume here and throughout that $X_i \in \mathbb{R}^{n \times p_i}$, $i = 1, \ldots, d$ each have full column rank so that the updates in (5) are uniquely defined (this is used for convenience, and is not a strong assumption; note that there is no restriction on the dimensionality of the full problem in (2), i.e., we could still have $X \in \mathbb{R}^{n \times p}$ with $p \gg n$). The precise form of these updates, of course, depends on the penalty functions. Suppose that each $h_i$ is the support function of a closed, convex set $D_i \subseteq \mathbb{R}^{p_i}$, i.e.,

$$
h_i(v) = \max_{d \in D_i} \langle d, v \rangle, \quad \text{for } i = 1, \ldots, d.
$$

Suppose also that $C_i = (X_i^T)^{-1}(D_i) = \{v \in \mathbb{R}^n : X_i^T v \in D_i\}$, the inverse image of $D_i$ under the linear map $X_i^T$, for $i = 1, \ldots, d$. Then, perhaps surprisingly, it turns out that the coordinate descent iterations (5) are exactly the same as the Dykstra iterations (4), via a duality argument. We extract the key relationship as a lemma below, for future reference, and then state the formal equivalence. Proofs of these results, as with all results in this paper, are given in the supplement.

**Lemma 1.** *Assume that $X_i \in \mathbb{R}^{n \times p_i}$ has full column rank and $h_i(v) = \max_{d \in D_i} \langle d, v \rangle$ for a closed, convex set $D_i \subseteq \mathbb{R}^{p_i}$. Then for $C_i = (X_i^T)^{-1}(D_i) \subseteq \mathbb{R}^n$ and any $b \in \mathbb{R}^n$,*

$$
\hat{w}_i = \operatorname*{argmin}_{w_i \in \mathbb{R}^{p_i}} \frac{1}{2} \| b - X_i w_i \|_2^2 + h_i(w_i) \iff X_i \hat{w}_i = (\mathrm{Id} - P_{C_i})(b).
$$

*where $\mathrm{Id}(\cdot)$ denotes the identity mapping.*

**Theorem 1.** *Assume the setup in Lemma 1, for each $i = 1, \ldots, d$. Then problems (1), (2) are dual to each other, and their solutions, denoted $\hat{u}, \hat{w}$, respectively, satisfy $\hat{u} = y - X\hat{w}$. Further, Dykstra's algorithm (4) and coordinate descent (5) are equivalent, and satisfy at all iterations $k = 1, 2, 3, \ldots$:*

$$
z_i^{(k)} = X_i w_i^{(k)} \quad \text{and} \quad u_i^{(k)} = y - \sum_{j \leq i} X_j w_j^{(k)} - \sum_{j > i} X_j w_j^{(k-1)}, \quad \text{for } i = 1, \ldots, d.
$$

The equivalence between coordinate descent and Dykstra's algorithm dates back to (at least) Han (1988); Gaffke and Mathar (1989), under the special case $X = I$. In fact, Han (1988), presumably unaware of Dykstra's algorithm, seems to have reinvented the method and established convergence

through its relationship to coordinate descent. This work then inspired Tseng (1993) (who must have also been unaware of Dykstra's algorithm) to improve the existing analyses of coordinate descent, which at the time all assumed smoothness of the objective function. (Tseng continued on to become arguably the single most important contributor to the theory of coordinate descent of the 1990s and 2000s, and his seminal work Tseng (2001) is still one of the most comprehensive analyses to date.)

References to this equivalence can be found speckled throughout the literature on Dykstra's method, but given the importance of the regularized problem form (2) for modern statistical and machine learning estimation tasks, we feel that the connection between Dykstra's algorithm and coordinate descent and is not well-known enough and should be better explored. In what follows, we show that some old work on Dykstra's algorithm, fed through this equivalence, yields new convergence results for coordinate descent for the lasso and a new parallel version of coordinate descent.

**ADMM.** The augmented Lagrangian method of multipliers or ADMM was invented by Glowinski and Marroco (1975); Gabay and Mercier (1976). ADMM is a member of a class of methods generally called *operator splitting techniques*, and is equivalent (via a duality argument) to *Douglas-Rachford splitting* (Douglas and Rachford, 1956; Lions and Mercier, 1979). Recently, there has been a strong revival of interest in ADMM (and operator splitting techniques in general), arguably due (at least in part) to the popular monograph of Boyd et al. (2011), where it is argued that the ADMM framework offers an appealing flexibility in algorithm design, which permits parallelization in many nontrivial situations. As with coordinate descent, it would be difficult thoroughly describe recent developments on ADMM, given the magnitude and pace of the literature on this topic. To give just a few examples, recent progress includes finite-time linear convergence rates for ADMM (see Nishihara et al. 2015; Hong and Luo 2017 and references therein), and accelerated extensions of ADMM (see Goldstein et al. 2014; Kadkhodaie et al. 2015 and references therein).

To derive an ADMM algorithm for (1), we introduce auxiliary variables and equality constraints to put the problem in a suitable ADMM form. While different formulations for the auxiliary variables and constraints give rise to different algorithms, loosely speaking, these algorithms generally take on similar forms to Dykstra's algorithm for (1). The same is also true of ADMM for the *set intersection problem*, a simpler task than the best approximation problem (1), in which we only seek a point in the intersection $C_1 \cap \cdots \cap C_d$, and solve

$$\min_{u \in \mathbb{R}^n} \ \sum_{i=1}^d I_{C_i}(u_i), \tag{6}$$

where $I_C(\cdot)$ denotes the indicator function of a set $C$ (equal to 0 on $C$, and $\infty$ otherwise). Consider the case of $d = 2$ sets, in which case the translation of (6) into ADMM form is unambiguous. ADMM for (6), properly initialized, appears highly similar to Dykstra's algorithm for (1); so similar, in fact, that Boyd et al. (2011) mistook the two algorithms for being equivalent, which is not generally true, and was shortly thereafter corrected by Bauschke and Koch (2013).

Below we show that when $d = 2$, $C_1$ is a linear subspace, and $y \in C_1$, an ADMM algorithm for (1) (and not the simpler set intersection problem (6)) is indeed equivalent to Dykstra's algorithm for (1). Introducing auxiliary variables, the problem (1) becomes

$$\min_{u_1, u_2 \in \mathbb{R}^n} \ \|y - u_1\|_2^2 + I_{C_1}(u_1) + I_{C_2}(u_2) \quad \text{subject to} \quad u_1 = u_2.$$

The augmented Lagrangian is $L(u_1, u_2, z) = \|y - u_1\|_2^2 + I_{C_1}(u_1) + I_{C_2}(u_2) + \rho\|u_1 - u_2 + z\|_2^2 - \rho\|z\|_2^2$, where $\rho > 0$ is an augmented Lagrangian parameter. ADMM repeats, for $k = 1, 2, 3, \ldots$:

$$\begin{aligned}
u_1^{(k)} &= P_{C_1}\left(\frac{y}{1+\rho} + \frac{\rho(u_2^{(k-1)} - z^{(k-1)})}{1+\rho}\right), \\
u_2^{(k)} &= P_{C_2}(u_1^{(k)} + z^{(k-1)}), \\
z^{(k)} &= z^{(k-1)} + u_1^{(k)} - u_2^{(k)}.
\end{aligned} \tag{7}$$

Suppose we initialize $u_2^{(0)} = y$, $z^{(0)} = 0$, and set $\rho = 1$. Using linearity of $P_{C_1}$, the fact that $y \in C_1$, and a simple inductive argument, the above iterations can be rewritten as

$$\begin{aligned}
u_1^{(k)} &= P_{C_1}(u_2^{(k-1)}), \\
u_2^{(k)} &= P_{C_2}(u_1^{(k)} + z^{(k-1)}), \\
z^{(k)} &= z^{(k-1)} + u_1^{(k)} - u_2^{(k)},
\end{aligned} \tag{8}$$

which is precisely the same as Dykstra's iterations (4), once we realize that, due again to linearity of $P_{C_1}$, the sequence $z_1^{(k)}$, $k = 1, 2, 3, \ldots$ in Dykstra's iterations plays no role and can be ignored.

Though $d = 2$ sets in (1) may seem like a rather special case, the strategy for parallelization in both Dykstra's algorithm and ADMM stems from rewriting a general $d$-set problem as a 2-set problem, so the above connection between Dykstra's algorithm and ADMM can be relevant even for problems with $d > 2$, and will reappear in our later discussion of parallel coordinate descent. As a matter of conceptual interest only, we note that for general $d$ (and no constraints on the sets being subspaces), Dykstra's iterations (4) can be viewed as a limiting version of the ADMM iterations either for (1) or for (6), as we send the augmented Lagrangian parameters to $\infty$ or to 0 at particular scalings. See the supplement for details.

## 3  Coordinate descent for the lasso

The *lasso* problem (Tibshirani, 1996; Chen et al., 1998), defined for a tuning parameter $\lambda \geq 0$ as

$$\min_{w \in \mathbb{R}^p} \frac{1}{2} \|y - Xw\|_2^2 + \lambda \|w\|_1, \tag{9}$$

is a special case of (2) where the coordinate blocks are of each size 1, so that $X_i \in \mathbb{R}^n$, $i = 1, \ldots, p$ are just the columns of $X$, and $w_i \in \mathbb{R}$, $i = 1, \ldots, p$ are the components of $w$. This problem fits into the framework of (2) with $h_i(w_i) = \lambda |w_i| = \max_{d \in D_i} dw_i$ for $D_i = [-\lambda, \lambda]$, for each $i = 1, \ldots, d$.

Coordinate descent is widely-used for the lasso (9), both because of the simplicity of the coordinate-wise updates, which reduce to soft-thresholding, and because careful implementations can achieve state-of-the-art performance, at the right problem sizes. The use of coordinate descent for the lasso was popularized by Friedman et al. (2007, 2010), but was studied earlier or concurrently by several others, e.g., Fu (1998); Sardy et al. (2000); Wu and Lange (2008).

As we know from Theorem 1, the dual of problem (9) is the best approximation problem (1), where $C_i = (X_i^T)^{-1}(D_i) = \{v \in \mathbb{R}^n : |X_i^T v| \leq \lambda\}$ is an intersection of two halfspaces, for $i = 1, \ldots, p$. This makes $C_1 \cap \cdots \cap C_d$ an intersection of $2p$ halfspaces, i.e., a (centrally symmetric) polyhedron. For projecting onto a polyhedron, it is well-known that Dykstra's algorithm reduces to *Hildreth's algorithm* (Hildreth, 1957), an older method for quadratic programming that itself has an interesting history in optimization. Theorem 1 hence shows coordinate descent for the lasso (9) is equivalent not only to Dykstra's algorithm, but also to Hildreth's algorithm, for (1).

This equivalence suggests a number of interesting directions to consider. For example, key practical speedups have been developed for coordinate descent for the lasso that enable this method to attain state-of-the-art performance at the right problem sizes, such as clever updating rules and screening rules (e.g., Friedman et al. 2010; El Ghaoui et al. 2012; Tibshirani et al. 2012; Wang et al. 2015). These implementation tricks can now be used with Dykstra's (Hildreth's) algorithm. On the flip side, as we show next, older results from Iusem and De Pierro (1990); Deutsch and Hundal (1994) on Dykstra's algorithm for polyhedra, lead to interesting new results on coordinate descent for the lasso.

**Theorem 2** (**Adaptation of Iusem and De Pierro 1990**). *Assume the columns of $X \in \mathbb{R}^{n \times p}$ are in general position, and $\lambda > 0$. Then coordinate descent for the lasso (9) has an asymptotically linear convergence rate, in that for large enough $k$,*

$$\frac{\|w^{(k+1)} - \hat{w}\|_\Sigma}{\|w^{(k)} - \hat{w}\|_\Sigma} \leq \left( \frac{a^2}{a^2 + \lambda_{\min}(X_A^T X_A) / \max_{i \in A} \|X_i\|_2^2} \right)^{1/2}, \tag{10}$$

*where $\hat{w}$ is the lasso solution in (9), $\Sigma = X^T X$, and $\|z\|_\Sigma^2 = z^T \Sigma z$ for $z \in \mathbb{R}^p$, $A = \operatorname{supp}(\hat{w})$ is the active set of $\hat{w}$, $a = |A|$ is its size, $X_A \in \mathbb{R}^{n \times a}$ denotes the columns of $X$ indexed by $A$, and $\lambda_{\min}(X_A^T X_A)$ denotes the smallest eigenvalue of $X_A^T X_A$.*

**Theorem 3** (**Adaptation of Deutsch and Hundal 1994**). *Assume the same conditions and notation as in Theorem 2. Then for large enough $k$,*

$$\frac{\|w^{(k+1)} - \hat{w}\|_\Sigma}{\|w^{(k)} - \hat{w}\|_\Sigma} \leq \left( 1 - \prod_{j=1}^{a-1} \frac{\|P_{\{i_{j+1}, \ldots, i_a\}}^\perp X_{i_j}\|_2^2}{\|X_{i_j}\|_2^2} \right)^{1/2}, \tag{11}$$

*where we enumerate $A = \{i_1, \ldots, i_a\}$, $i_1 < \ldots < i_a$, and we denote by $P_{\{i_{j+1}, \ldots, i_a\}}^\perp$ the projection onto the orthocomplement of the column span of $X_{\{i_{j+1}, \ldots, i_a\}}$.*

The results in Theorems 2, 3 both rely on the assumption of general position for the columns of $X$. This is only used for convenience and can be removed at the expense of more complicated notation. Loosely put, the general position condition simply rules out trivial linear dependencies between small numbers of columns of $X$, but places no restriction on the dimensions of $X$ (i.e., it still allows for $p \gg n$). It implies that the lasso solution $\hat{w}$ is unique, and that $X_A$ (where $A = \text{supp}(\hat{w})$) has full column rank. See Tibshirani (2013) for a precise definition of general position and proofs of these facts. We note that when $X_A$ has full column rank, the bounds in (10), (11) are strictly less than 1.

**Remark 1** (**Comparing** (10) **and** (11)). Clearly, both the bounds in (10), (11) are adversely affected by correlations among $X_i$, $i \in A$ (i.e., stronger correlations will bring each closer to 1). It seems to us that (11) is usually the smaller of the two bounds, based on simple mathematical and numerical comparisons. More detailed comparisons would be interesting, but is beyond the scope of this paper.

**Remark 2** (**Linear convergence without strong convexity**). One striking feature of the results in Theorems 2, 3 is that they guarantee (asymptotically) linear convergence of the coordinate descent iterates for the lasso, with no assumption about strong convexity of the objective. More precisely, there are no restrictions on the dimensionality of $X$, so we enjoy linear convergence *even without an assumption on the smooth part of the objective*. This is in line with classical results on coordinate descent for smooth functions, see, e.g., Luo and Tseng (1992). The modern finite-time convergence analyses of coordinate descent do not, as far as we understand, replicate this remarkable property. For example, Beck and Tetruashvili (2013); Li et al. (2016) establish finite-time linear convergence rates for coordinate descent, but require strong convexity of the entire objective.

**Remark 3** (**Active set identification**). The asymptotics developed in Iusem and De Pierro (1990); Deutsch and Hundal (1994) are based on a notion of (in)active set identification: the critical value of $k$ after which (10), (11) hold is based on the (provably finite) iteration number at which Dykstra's algorithm identifies the inactive halfspaces, i.e., at which coordinate descent identifies the inactive set of variables, $A^c = \text{supp}(\hat{w})^c$. This might help explain why in practice coordinate descent for the lasso performs exceptionally well with warm starts, over a decreasing sequence of tuning parameter values $\lambda$ (e.g., Friedman et al. 2007, 2010): here, each coordinate descent run is likely to identify the (in)active set—and hence enter the linear convergence phase—at an early iteration number.

## 4 Parallel coordinate descent

**Parallel-Dykstra-CD.** An important consequence of the connection between Dykstra's algorithm and coordinate descent is a new parallel version of the latter, stemming from an old parallel version of the former. A parallel version of Dykstra's algorithm is usually credited to Iusem and Pierro (1987) for polyhedra and Gaffke and Mathar (1989) for general sets, but really the idea dates back to the product space formalization of Pierra (1984). We rewrite problem (1) as

$$\min_{u=(u_1,\ldots,u_d)\in\mathbb{R}^{nd}} \sum_{i=1}^{d} \gamma_i \|y - u_i\|_2^2 \quad \text{subject to} \quad u \in C_0 \cap (C_1 \times \cdots \times C_d), \qquad (12)$$

where $C_0 = \{(u_1,\ldots,u_d) \in \mathbb{R}^{nd} : u_1 = \cdots = u_d\}$, and $\gamma_1,\ldots,\gamma_d > 0$ are weights that sum to 1. After rescaling appropriately to turn (12) into an unweighted best approximation problem, we can apply Dykstra's algorithm, which sets $u_1^{(0)} = \cdots = u_d^{(0)} = y$, $z_1^{(0)} = \cdots = z_d^{(0)} = 0$, and repeats:

$$
\begin{aligned}
u_0^{(k)} &= \sum_{i=1}^{d} \gamma_i u_i^{(k-1)}, \\
u_i^{(k)} &= P_{C_i}(u_0^{(k)} + z_i^{(k-1)}), \\
z_i^{(k)} &= u_0^{(k)} + z_i^{(k-1)} - u_i^{(k)},
\end{aligned}
\left.\begin{aligned}\\ \\ \\ \end{aligned}\right\} \quad \text{for } i = 1,\ldots,d,
\qquad (13)
$$

for $k = 1, 2, 3, \ldots$. The steps enclosed in curly brace above can all be performed in parallel, so that (13) is a parallel version of Dykstra's algorithm (4) for (1). Applying Lemma 1, and a straightforward inductive argument, the above algorithm can be rewritten as follows. We set $w^{(0)} = 0$, and repeat:

$$w_i^{(k)} = \operatorname*{argmin}_{w_i \in \mathbb{R}^{p_i}} \frac{1}{2}\left\|y - Xw^{(k-1)} + X_i w_i^{(k-1)}/\gamma_i - X_i w_i/\gamma_i\right\|_2^2 + h_i(w_i/\gamma_i), \quad i = 1,\ldots,d, \quad (14)$$

for $k = 1, 2, 3, \ldots$, which we call *parallel-Dykstra-CD* (with CD being short for coordinate descent). Again, note that the each of the $d$ coordinate updates in (14) can be performed in parallel, so that

(14) is a parallel version of coordinate descent (5) for (2). Also, as (14) is just a reparametrization of Dykstra's algorithm (13) for the 2-set problem (12), it is guaranteed to converge in full generality, as per the standard results on Dykstra's algorithm (Han, 1988; Gaffke and Mathar, 1989).

**Theorem 4.** *Assume that $X_i \in \mathbb{R}^{n \times p_i}$ has full column rank and $h_i(v) = \max_{d \in D_i} \langle d, v \rangle$ for a closed, convex set $D_i \subseteq \mathbb{R}^{p_i}$, for $i = 1, \ldots, d$. If (2) has a unique solution, then the iterates in (14) converge to this solution. More generally, if the interior of $\cap_{i=1}^{d} (X_i^T)^{-1}(D_i)$ is nonempty, then the sequence $w^{(k)}$, $k = 1, 2, 3, \ldots$ from (14) has at least one accumulation point, and any such point solves (2). Further, $Xw^{(k)}$, $k = 1, 2, 3, \ldots$ converges to $X\hat{w}$, the optimal fitted value in (2).*

There have been many recent exciting contributions to the parallel coordinate descent literature; two standouts are Jaggi et al. (2014); Richtarik and Takac (2016), and numerous others are described in Wright (2015). What sets parallel-Dykstra-CD apart, perhaps, is its simplicity: convergence of the iterations (14), given in Theorem 4, just stems from the connection between coordinate descent and Dykstra's algorithm, and the fact that the parallel Dykstra iterations (13) are nothing more than the usual Dykstra iterations after a product space reformulation. Moreover, parallel-Dykstra-CD for the lasso enjoys an (asymptotic) linear convergence rate under essentially no assumptions, thanks once again to an old result on the parallel Dykstra (Hildreth) algorithm from Iusem and De Pierro (1990). The details can be found in the supplement.

**Parallel-ADMM-CD.**   As an alternative to the parallel method derived using Dykstra's algorithm, ADMM can also offer a version of parallel coordinate descent. Since (12) is a best approximation problem with $d = 2$ sets, we can refer back to our earlier ADMM algorithm in (7) for this problem. By passing these ADMM iterations through the connection developed in Lemma 1, we arrive at what we call *parallel-ADMM-CD*, which initializes $u_0^{(0)} = y$, $w^{(-1)} = w^{(0)} = 0$, and repeats:

$$
\begin{aligned}
u_0^{(k)} &= \frac{(\sum_{i=1}^{d} \rho_i) u_0^{(k-1)}}{1 + \sum_{i=1}^{d} \rho_i} + \frac{y - Xw^{(k-1)}}{1 + \sum_{i=1}^{d} \rho_i} + \frac{X(w^{(k-2)} - w^{(k-1)})}{1 + \sum_{i=1}^{d} \rho_i}, \\
w_i^{(k)} &= \operatorname*{argmin}_{w_i \in \mathbb{R}^{p_i}} \frac{1}{2} \left\| u_0^{(k)} + X_i w_i^{(k-1)}/\rho_i - X_i w_i/\rho_i \right\|_2^2 + h_i(w_i/\rho_i), \quad i = 1, \ldots, d,
\end{aligned}
\tag{15}
$$

for $k = 1, 2, 3, \ldots$, where $\rho_1, \ldots, \rho_d > 0$ are augmented Lagrangian parameters. In each iteration, the updates to $w_i^{(k)}$, $i = 1, \ldots, d$ above can be done in parallel. Just based on their form, it seems that (15) can be seen as a parallel version of coordinate descent (5) for problem (2). The next result confirms this, leveraging standard theory for ADMM (Gabay, 1983; Eckstein and Bertsekas, 1992).

**Theorem 5.** *Assume that $X_i \in \mathbb{R}^{n \times p_i}$ has full column rank and $h_i(v) = \max_{d \in D_i} \langle d, v \rangle$ for a closed, convex set $D_i \subseteq \mathbb{R}^{p_i}$, for $i = 1, \ldots, d$. Then the sequence $w^{(k)}$, $k = 1, 2, 3, \ldots$ in (15) converges to a solution in (2).*

The parallel-ADMM-CD iterations in (15) and parallel-Dykstra-CD iterations in (14) differ in that, where the latter uses a residual $y - Xw^{(k-1)}$, the former uses an iterate $u_0^{(k)}$ that seems to have a more complicated form, being a convex combination of $u_0^{(k-1)}$ and $y - Xw^{(k-1)}$, plus a quantity that acts like a momentum term. It turns out that when $\rho_1, \ldots, \rho_d$ sum to 1, the two methods (14), (15) are exactly the same. While this may seem like a surprising coincidence, it is in fact nothing more than a reincarnation of the previously established equivalence between Dykstra's algorithm (4) and ADMM (8) for a 2-set best approximation problem, as here $C_0$ is a linear subspace.

Of course, with ADMM we need not choose probability weights for $\rho_1, \ldots, \rho_d$, and the convergence in Theorem 5 is guaranteed for any fixed values of these parameters. Thus, even though they were derived from different perspectives, parallel-ADMM-CD subsumes parallel-Dykstra-CD, and it is a strictly more general approach. It is important to note that larger values of $\rho_1, \ldots, \rho_d$ can often lead to faster convergence in practice, as we show in Figure 1. More detailed study and comparisons to related parallel methods are worthwhile, but are beyond the scope of this work.

## 5   Discussion and extensions

We studied connections between Dykstra's algorithm, ADMM, and coordinate descent. Leveraging these connections, we established an (asymptotically) linear convergence rate for coordinate descent for the lasso, as well as two parallel versions of coordinate descent (one based on Dykstra's algorithm and the other on ADMM). Some extensions and possibilities for future work are described below.

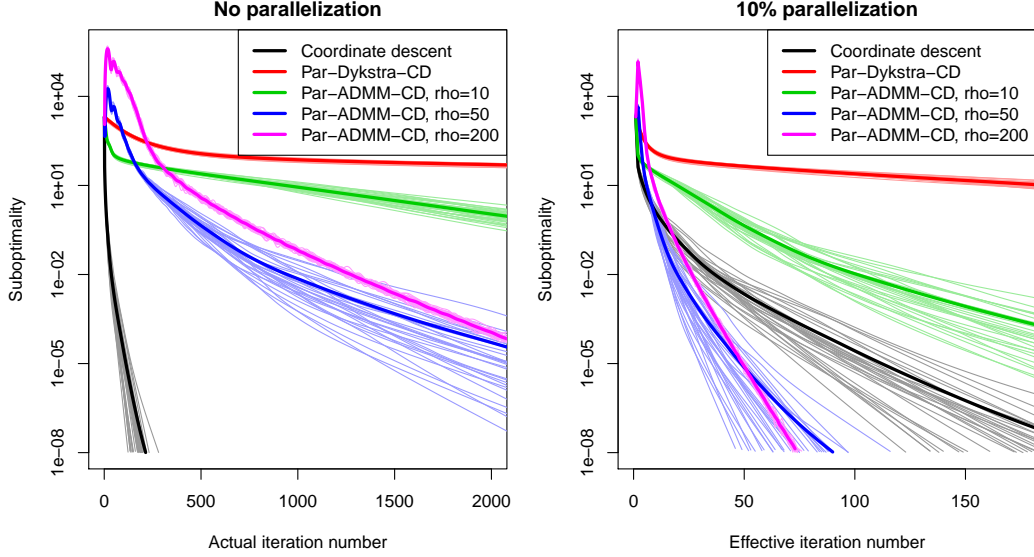

Figure 1: *Suboptimality curves for serial coordinate descent, parallel-Dykstra-CD, and three tunings of parallel-ADMM-CD (i.e., three different values of $\rho = \sum_{i=1}^{p} \rho_i$), each run over the same 30 lasso problems with $n = 200$ and $p = 500$. For details of the experimental setup, see the supplement.*

**Nonquadratic loss: Dykstra's algorithm and coordinate descent.** Given a convex function $f$, a generalization of (2) is the *regularized estimation problem*

$$\min_{w \in \mathbb{R}^p} f(Xw) + \sum_{i=1}^{d} h_i(w_i). \tag{16}$$

Regularized regression (2) is given by $f(z) = \frac{1}{2}\|y - z\|_2^2$, and e.g., regularized classification (under the logistic loss) by $f(z) = -y^T z + \sum_{i=1}^{n} \log(1 + e^{z_i})$. In (block) coordinate descent for (16), we initialize say $w^{(0)} = 0$, and repeat, for $k = 1, 2, 3, \ldots$:

$$w_i^{(k)} = \underset{w_i \in \mathbb{R}^{p_i}}{\operatorname{argmin}} \, f\left(\sum_{j<i} X_j w_j^{(k)} + \sum_{j>i} X_j w_j^{(k-1)} + X_i w_i\right) + h_i(w_i), \quad i = 1, \ldots, d. \tag{17}$$

On the other hand, given a differentiable and strictly convex function $g$, we can generalize (1) to the following *best Bregman-approximation problem*,

$$\min_{u \in \mathbb{R}^n} \, D_g(u, b) \quad \text{subject to} \quad u \in C_1 \cap \cdots \cap C_d. \tag{18}$$

where $D_g(u, b) = g(u) - g(b) - \langle \nabla g(b), u - b \rangle$ is the *Bregman divergence* between $u$ and $b$ with respect to $g$. When $g(v) = \frac{1}{2}\|v\|_2^2$ (and $b = y$), this recovers the best approximation problem (1). As shown in Censor and Reich (1998); Bauschke and Lewis (2000), Dykstra's algorithm can be extended to apply to (18). We initialize $u_d^{(0)} = b$, $z_1^{(0)} = \cdots = z_d^{(0)} = 0$, and repeat for $k = 1, 2, 3, \ldots$:

$$\left.\begin{aligned} u_0^{(k)} &= u_d^{(k-1)}, \\ u_i^{(k)} &= (P_{C_i}^g \circ \nabla g^*)\left(\nabla g(u_{i-1}^{(k)}) + z_i^{(k-1)}\right), \\ z_i^{(k)} &= \nabla g(u_{i-1}^{(k)}) + z_i^{(k-1)} - \nabla g(u_i^{(k)}), \end{aligned}\right\} \quad \text{for } i = 1, \ldots, d, \tag{19}$$

where $P_C^g(x) = \operatorname{argmin}_{c \in C} D_g(c, x)$ denotes the Bregman (rather than Euclidean) projection of $x$ onto a set $C$, and $g^*$ is the conjugate function of $g$. Though it may not be immediately obvious, when $g(v) = \frac{1}{2}\|v\|_2^2$ the above iterations (19) reduce to the standard (Euclidean) Dykstra iterations in (4). Furthermore, Dykstra's algorithm and coordinate descent are equivalent in the more general setting.

**Theorem 6.** *Let $f$ be a strictly convex, differentiable function that has full domain. Assume that $X_i \in \mathbb{R}^{n \times p_i}$ has full column rank and $h_i(v) = \max_{d \in D_i} \langle d, v \rangle$ for a closed, convex set $D_i \subseteq \mathbb{R}^{p_i}$, for $i = 1, \ldots, d$. Also, let $g(v) = f^*(-v)$, $b = -\nabla f(0)$, and $C_i = (X_i^T)^{-1}(D_i) \subseteq \mathbb{R}^n$, $i = 1, \ldots, d$.*

*Then* (16), (18) *are dual to each other, and their solutions* $\hat{w}, \hat{u}$ *satisfy* $\hat{u} = -\nabla f(X\hat{w})$. *Moreover, Dykstra's algorithm* (19) *and coordinate descent* (17) *are equivalent, i.e., for* $k = 1, 2, 3, \ldots$:

$$z_i^{(k)} = X_i w_i^{(k)} \quad and \quad u_i^{(k)} = -\nabla f\left(\sum_{j \leq i} X_j w_j^{(k)} + \sum_{j > i} X_j w_j^{(k-1)}\right), \quad for \ i = 1, \ldots, d.$$

**Nonquadratic loss: parallel coordinate descent methods.** For a general regularized estimation problem (16), parallel coordinate descent methods can be derived by applying Dykstra's algorithm and ADMM to a product space reformulation of the dual. Interestingly, the subsequent coordinate descent algorithms are *no longer equivalent* (for a unity augmented Lagrangian parameter), and they feature quite different computational structures. Parallel-Dykstra-CD for (16) initializes $w^{(0)} = 0$, and repeats:

$$w_i^{(k)} = \underset{w_i \in \mathbb{R}^{p_i}}{\operatorname{argmin}} \ f\left(Xw^{(k)} - X_i w_i^{(k)}/\gamma_i + X_i w_i/\gamma_i\right) + h_i(w_i/\gamma_i), \quad i = 1, \ldots, d, \qquad (20)$$

for $k = 1, 2, 3, \ldots$, and weights $\gamma_1, \ldots, \gamma_d > 0$ that sum to 1. In comparison, parallel-ADMM-CD for (16) begins with $u_0^{(0)} = 0$, $w^{(-1)} = w^{(0)} = 0$, and repeats:

$$\text{Find } u_0^{(k)} \text{ such that:} \quad u_0^{(k)} = -\nabla f\left(\left(\sum_{i=1}^d \rho_i\right)(u_0^{(k)} - u_0^{(k-1)}) - X(w^{(k-2)} - 2w^{(k-1)})\right),$$

$$w_i^{(k)} = \underset{w_i \in \mathbb{R}^{p_i}}{\operatorname{argmin}} \ \frac{1}{2}\left\|u_0^{(k)} + X_i w_i^{(k-1)}/\rho_i - X_i w_i/\rho_i\right\|_2^2 + h_i(w_i/\rho_i), \quad i = 1, \ldots, d, \qquad (21)$$

for $k = 1, 2, 3, \ldots$, and parameters $\rho_1, \ldots, \rho_d > 0$. Derivation details are given in the supplement. Notice the stark contrast between the parallel-Dykstra-CD iterations (20) and the parallel-ADMM-CD iterations (21). In (20), we perform (in parallel) coordinatewise $h_i$-regularized minimizations involving $f$, for $i = 1, \ldots, d$. In (21), we perform a single quadratically-regularized minimization involving $f$ for the $u_0$-update, and then for the $w$-update, we perform (in parallel) coordinatewise $h_i$-regularized minimizations involving a quadratic loss, for $i = 1, \ldots, d$ (these are typically much cheaper than the analogous minimizations for typical nonquadratic losses $f$ of interest).

We note that the $u_0$-update in the parallel-ADMM-CD iterations (21) simplifies for many losses $f$ of interest; in particular, for separable loss functions of the form $f(v) = \sum_{i=1}^n f_i(v_i)$, for convex, univariate functions $f_i$, $i = 1, \ldots, n$, the $u_0$-update separates into $n$ univariate minimizations. As an example, consider the *logistic lasso* problem,

$$\min_{w \in \mathbb{R}^p} \ -y^T X w + \sum_{i=1}^n \log(1 + e^{x_i^T w}) + \lambda\|w\|_1, \qquad (22)$$

where $x_i \in \mathbb{R}^p$, $i = 1, \ldots, n$ denote the rows of $X$. Abbreviating $\rho = \sum_{i=1}^p \rho_i$, and denoting by $\sigma(x) = 1/(1 + e^{-x})$ the sigmoid function, and by $S_t(x) = \operatorname{sign}(x)(|x| - t)_+$ the soft-thresholding function at a level $t > 0$, the parallel-ADMM-CD iterations (21) for (22) reduce to:

$$\text{Find } u_{0i}^{(k)} \text{ such that:} \quad u_{0i}^{(k)} = y_i - \sigma(\rho u_{0i}^{(k)} - c_i^{(k)}), \quad i = 1, \ldots, n,$$

$$w_i^{(k)} = S_{\lambda\rho_i/\|X_i\|_2^2}\left(\frac{\rho_i X_i^T(u_0^{(k)} + X_i w_i^{(k-1)}/\rho_i)}{\|X_i\|_2^2}\right), \quad i = 1, \ldots, p, \qquad (23)$$

where $c_i^{(k)} = \rho u_{0i}^{(k-1)} + x_i^T(w^{(k-2)} - 2w^{(k-1)})$, for $i = 1, \ldots, n$, $k = 1, 2, 3, \ldots$. We see that both the $u_0$-update and $w$-update in (23) can be parallelized, and each coordinate update in the former can be done with, say, a simple bisection search.

**Asynchronous parallel algorithms, and coordinate descent in Hilbert spaces.** We finish with some directions for possible future work. Asynchronous variants of parallel coordinate descent are currently of great interest, e.g., see the review in Wright (2015). Given the link between ADMM and coordinate descent developed in this paper, it would be interesting to investigate the implications of the recent exciting progress on asynchronous ADMM, e.g., see Chang et al. (2016a,b) and references therein, for coordinate descent. In a separate direction, much of the literature on Dykstra's algorithm emphasizes that this method works seamlessly in Hilbert spaces. It would be interesting to consider the connections to (parallel) coordinate descent in infinite-dimensional function spaces, which we would encounter, e.g., in alternating conditional expectation algorithms or backfitting algorithms in additive models.

## Footnotes

[1]To be precise, this is *cyclic* coordinate descent, where *exact* minimization is performed along each block of coordinates. *Randomized* versions of this algorithm have recently become popular, as have *inexact* or *proximal* versions. While these variants are interesting, they are not the focus of our paper.

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
