[Supplementary Material]

# Supplement to: "Dykstra's Algorithm, ADMM, and Coordinate Descent: Connections, Insights, and Extensions"

**Ryan J. Tibshirani**
Department of Statistics and Machine Learning Department
Carnegie Mellon University
Pittsburgh, PA 15213
ryantibs@stat.cmu.edu

We give additional details and proofs for the results in "Dykstra's Algorithm, ADMM, and Coordinate Descent: Connections, Insights, and Extensions".

## A.1 Proofs of Lemma 1 and Theorem 1

These results are direct consequences of the more general Lemma A.2 and Theorem 6, respectively, when $f(v) = \frac{1}{2}\|y - v\|_2^2$ (and so $f^*(v) = -\frac{1}{2}\|y\|_2^2 + \frac{1}{2}\|y + v\|_2^2$); see Section A.9 below for their proofs.

## A.2 Dykstra's algorithm and ADMM for the $d$-set best approximation and set intersection problems

Here we show that, under an inertial-type modification, the ADMM iterations for (6) are in a certain limiting sense equivalent to Dykstra's iterations for (1). We introduce auxiliary variables to transform problem (6) into

$$\min_{u_0,\ldots,u_d \in \mathbb{R}^n} \sum_{i=1}^{d} I_{C_i}(u) \quad \text{subject to} \quad u_d = u_0, u_0 = u_1, \ldots, u_{d-1} = u_d,$$

and the corresponding augmented Lagrangian is $L(u_0, \ldots, u_d, z_0, \ldots, z_d) = \rho_0 \|u_d - u_0 + z_0\|_2^2 + \sum_{i=1}^{d}(I_{C_i}(u) + \rho_i\|u_{i-1} - u_i + z_i\|_2^2) - \sum_{i=0}^{d} \rho_i\|z_i\|_2^2$, with $\rho_0, \ldots, \rho_d > 0$ being augmented Lagrangian parameters. ADMM is defined by repeating the updates:

$$u_i^{(k)} = \operatorname*{argmin}_{u_i \in \mathbb{R}^n} L(u_0^{(k)}, \ldots, u_{i-1}^{(k)}, u_i, u_{i+1}^{(k-1)}, \ldots, u_d^{(k-1)}), \quad i = 0, \ldots, d,$$

$$z_i^{(k)} = z_i^{(k-1)} + u_{i-1}^{(k)} - u_i^{(k)}, \quad i = 0, \ldots, d,$$

for $k = 1, 2, 3, \ldots$, where we use $u_{-1}^{(k)} = u_d^{(k)}$ for convenience. Now consider an inertial modification in which, for the $u_0$ update above, we add the term $\|u_0 - u_d^{(k-1)}\|_2$ to the augmented Lagrangian in the minimization. A straightforward derivation then leads to the ADMM updates:

$$u_0^{(k)} = \frac{u_d^{(k-1)} + \rho_0(u_d^{(k-1)} + z_0^{(k-1)}) + \rho_1(u_1^{(k-1)} - z_1^{(k-1)})}{1 + \rho_0 + \rho_1},$$

$$u_i^{(k)} = P_{C_i}\left( \frac{(u_{i-1}^{(k)} + z_i^{(k-1)})}{1 + \rho_{i+1}/\rho_i} + \frac{(\rho_{i+1}/\rho_i)(u_{i+1}^{(k-1)} - z_{i+1}^{(k-1)})}{1 + \rho_{i+1}/\rho_i} \right), \quad i = 1, \ldots, d-1,$$

$$u_d^{(k)} = P_{C_d}\left( \frac{(u_{d-1}^{(k)} + z_d^{(k-1)})}{1 + \rho_0/\rho_d} + \frac{(\rho_0/\rho_d)(u_0^{(k)} - z_0^{(k-1)})}{1 + \rho_0/\rho_d} \right), \quad \text{(A.1)}$$

$$z_i^{(k)} = z_i^{(k-1)} + u_{i-1}^{(k)} - u_i^{(k)}, \quad i = 0, \ldots, d,$$

for $k = 1, 2, 3, \ldots$. Under the choices $\rho_0 = \alpha^{d+1}$ and $\rho_i = \alpha^i$, $i = 1, \ldots, d$, we see that as $\alpha \to 0$ the ADMM iterations (A.1) exactly coincide with the Dykstra iterations (4). Thus, under the proper initializations, $u_d^{(0)} = y$ and $z_0^{(0)} = \cdots = z_d^{(0)} = 0$, the limiting ADMM algorithm for (6) matches Dykstra's algorithm for (1).

Similar arguments can be used equate ADMM for (1) to Dykstra's algorithm, again in limiting sense. We rewrite (1) as

$$\min_{u_0, \ldots, u_d \in \mathbb{R}^n} \|y - u_0\|_2^2 + \sum_{i=1}^d I_{C_i}(u) \quad \text{subject to} \quad u_d = u_0, u_0 = u_1, \ldots, u_{d-1} = u_d.$$

Using an inertial modification for the $u_0$ update, where we now add the term $\rho_{-1} \|u_0 - u_d^{(k-1)}\|_2^2$ to the augmented Lagrangian in the minimization, the ADMM updates become:

$$u_0^{(k)} = \frac{y + \rho_{-1} u_d^{(k-1)} + \rho_0 (u_d^{(k-1)} + z_0^{(k-1)}) + \rho_1 (u_1^{(k-1)} - z_1^{(k-1)})}{1 + \rho_{-1} + \rho_0 + \rho_1},$$

$$u_i^{(k)} = P_{C_i} \left( \frac{(u_{i-1}^{(k)} + z_i^{(k-1)})}{1 + \rho_{i+1}/\rho_i} + \frac{(\rho_{i+1}/\rho_i)(u_{i+1}^{(k-1)} - z_{i+1}^{(k-1)})}{1 + \rho_{i+1}/\rho_i} \right), \quad i = 1, \ldots, d-1,$$

$$u_d^{(k)} = P_{C_d} \left( \frac{(u_{d-1}^{(k)} + z_d^{(k-1)})}{1 + \rho_0/\rho_d} + \frac{(\rho_0/\rho_d)(u_0^{(k)} - z_0^{(k-1)})}{1 + \rho_0/\rho_d} \right),$$

$$z_i^{(k)} = z_i^{(k-1)} + u_{i-1}^{(k)} - u_i^{(k)}, \quad i = 0, \ldots, d,$$

for $k = 1, 2, 3, \ldots$. Setting $\rho_{-1} = \alpha^{d+1}$, $\rho_0 = 1$, and $\rho_i = \alpha^{d+1-i}$, $i = 1, \ldots, d$, we can see that as $\alpha \to \infty$, the ADMM iterations (A.2) converge to the Dykstra iterations (4), and therefore with initializations $u_d^{(0)} = y$ and $z_0^{(0)} = \cdots = z_d^{(0)} = 0$, the limiting ADMM algorithm for (1) coincides with Dykstra's algorithm for the same problem.

The links above between ADMM and Dykstra's algorithm are intended to be of conceptual interest, and the ADMM algorithms (A.1), (A.2) may not be practically useful for arbitrary configurations of the augmented Lagrangian parameters. After all, both of these are multi-block ADMM approaches, and multi-block ADMM has subtle convergence behavior as studied, e.g., in Lin et al. (2015); Chen et al. (2016).

## A.3  Proof of Theorem 2

By Theorem 1, we know that coordinate descent applied to the lasso problem (9) is equivalent to Dykstra's algorithm on the best approximation problem (1), with $C_i = \{v \in \mathbb{R}^n : |X_i^T v| \leq \lambda\}$, for $i = 1, \ldots, p$. In particular, at the end of the $k$th iteration, it holds that

$$u_p^{(k)} = y - Xw^{(k)}, \quad \text{for } k = 1, 2, 3, \ldots.$$

By duality, we also have $\hat{u} = y - X\hat{w}$ at the solutions $\hat{u}, \hat{w}$ in (1), (9), respectively. Therefore any statement about the convergence of Dykstra's iterates may be translated into a statement about the convergence of the coordinate descent iterates, via the relationship

$$\|u^{(k)} - \hat{u}\|_2 = \|Xw^{(k)} - X\hat{w}\|_2 = \|w^{(k)} - \hat{w}\|_\Sigma, \quad \text{for } k = 1, 2, 3, \ldots. \tag{A.3}$$

We seek to apply the main result from Iusem and De Pierro (1990), on the asymptotic convergence rate of Dykstra's (Hildreth's) algorithm for projecting onto a polyhedron. One slight complication is that, in the current paramterization, coordinate descent is equivalent to Dykstra's algorithm on

$$C_1 \cap \ldots \cap C_p = \bigcap_{i=1}^p \{v \in \mathbb{R}^n : |X_i^T v| \leq \lambda\},$$

While polyhedral, the above is not explicitly an intersection of halfspaces (it is an intersection of slabs), which is the setup required by the analysis of Iusem and De Pierro (1990). Of course, we can simply define $C_i^+ = \{v \in \mathbb{R}^n : X_i^T v \leq \lambda\}$ and $C_i^- = \{v \in \mathbb{R}^n : X_i^T v \geq -\lambda\}$, $i = 1, \ldots, p$, and then the above intersection is equivalent to

$$C_1^+ \cap C_1^- \cap \ldots \cap C_p^+ \cap C_p^- = \bigcap_{i=1}^p \left( \{v \in \mathbb{R}^n : X_i^T v \leq \lambda\} \cap \{v \in \mathbb{R}^n : X_i^T v \geq -\lambda\} \right).$$

Moreover, one can check that the iterates from Dykstra's algorithm on $C_1^+ \cap C_1^- \cap \ldots \cap C_p^+ \cap C_p^-$ match[1] those from Dykstra's algorithm on $C_1 \cap \ldots \cap C_p$, provided that the algorithms cycle over the sets in the order they are written in these intersections. This means that the analysis of Iusem and De Pierro (1990) can be applied to coordinate descent for the lasso.

The error constant from Theorem 1 in Iusem and De Pierro (1990) is based on a geometric quantity that we explicitly lower bound below. It is not clear to us whether our lower bound is the best possible, and a better lower bound would improve the error constant presented in Theorem 2.

**Lemma A.1.** *Let $H_i = \{x \in \mathbb{R}^n : h_i^T x = b_i\}$, $i = 1, \ldots, s$ be hyperplanes, and $S = H_1 \cap \ldots \cap H_s$ the $s$-dimensional affine subspace formed by their intersection. For each $x \in \mathbb{R}^n$, denote by $H_x$ the hyperplane among $H_1, \ldots, H_s$ farthest from $x$. Define*

$$\mu = \inf_{x \in \mathbb{R}^n} \frac{d(x, H_x)}{d(x, S)},$$

*where $d(x, S) = \inf_{y \in S} \|x - y\|_2$ is the distance between $x$ and $S$, and similarly for $d(x, H_x)$. Then*

$$\mu \geq \frac{\sigma_{\min}(M)}{\sqrt{s} \max_{i=1,\ldots,s} \|h_i\|_2} > 0,$$

*where $M \in \mathbb{R}^{n \times s}$ has columns $h_1, \ldots, h_s$, and $\sigma_{\min}(M)$ is its smallest nonzero singular value.*

*Proof.* For any $x \in \mathbb{R}^n$, note that $d(x, S) = \|M^+(b - M^T x)\|_2$, where $M^+$ is the Moore-Penrose pseudoinverse of $M$. Also, $d(x, H_x) = \max_{i=1,\ldots,s} |b_i - h_i^T x|/\|h_i\|_2$. Hence, writing $\sigma_{\max}(M^+)$ for the maximum singular value of $M^+$,

$$\begin{aligned}
\frac{d(x, H_x)}{d(x, S)} &\geq \frac{\max_{i=1,\ldots,s} |b_i - h_i^T x|/\|h_i\|_2}{\sigma_{\max}(M^+)\|b - M^T x\|_2} \\
&\geq \frac{\sigma_{\min}(M)}{\max_{i=1,\ldots,s} \|h_i\|_2} \frac{\max_{i=1,\ldots,s} |b_i - h_i^T x|}{\|b - M^T x\|_2} \\
&\geq \frac{\sigma_{\min}(M)}{\sqrt{s} \max_{i=1,\ldots,s} \|h_i\|_2},
\end{aligned}$$

where we have used the fact that $\sigma_{\max}(M^+) = 1/\sigma_{\min}(M)$, as well as $\|v\|_\infty/\|v\|_2 \geq 1/\sqrt{s}$ for all vectors $v \in \mathbb{R}^s$. Taking an infimum over $x \in \mathbb{R}^n$ establishes the result. $\square$

Now we adapt and refine the result in Theorem 1 from Iusem and De Pierro (1990). These authors show that for large enough $k$,

$$\frac{\|u_p^{(k+1)} - \hat{u}\|_2}{\|u_p^{(k)} - \hat{u}\|_2} \leq \left(\frac{1}{1 + \sigma}\right)^{1/2},$$

where $\sigma = \mu^2/p$, and $\mu > 0$ is defined as follows. Let $A = \{i \in \{1, \ldots, p\} : |X_i^T \hat{u}| = \lambda\}$, and let $\rho = \text{sign}(X_A^T \hat{u})$. Also let

$$H_i = \{v \in \mathbb{R}^n : X_i^T v = \rho_i \lambda\}, \quad i \in A,$$

as well as $S = \cap_{i \in A} H_i$. Then

$$\mu = \inf_{x \in \mathbb{R}^n} \frac{d(x, H_x)}{d(x, S)},$$

where for each $x \in \mathbb{R}^n$, we denote by $H_x$ the hyperplane among $H_i$, $i \in A$ farthest from $x$.

In the nomenclature of the lasso problem, the set $A$ here is known as the *equicorrelation set*. The general position assumption on $X$ implies that the lasso $\hat{w}$ solution is unique, and that (for almost every in $y \in \mathbb{R}^n$), the equicorrelation set and support of $\hat{w}$ are equal, so we can write $A = \text{supp}(\hat{w})$. See Tibshirani (2013).

From Lemma A.1, we have that $\mu^2 \geq \lambda_{\min}(X_A^T X_A)/(a \max_{i \in A} \|X_i\|_2^2)$, where $a = |A|$, and so

$$\left(\frac{1}{1+\sigma}\right)^{1/2} \leq \left(\frac{pa}{pa + \lambda_{\min}(X_A^T X_A)/\max_{i \in A} \|X_i\|_2^2}\right)^{1/2}.$$

This is almost the desired result, but it is weaker, because of its dependence on $pa$ rather than $a^2$. Careful inspection of the proof of Theorem 1 in Iusem and De Pierro (1990) shows that the factor of $p$ in the constant $\sigma = \mu^2/p$ comes from an application of Cauchy-Schwartz, to derive an upper bound of the form (translated to our notation):

$$\left(\sum_{i=1}^{p-1} \|u_{i+1}^{(k)} - u_i^{(k)}\|_2\right)^2 \leq p \sum_{i=1}^{p-1} \|u_{i+1}^{(k)} - u_i^{(k)}\|_2^2.$$

See their equation (33) (in which, we note, there is a typo: the entire summation should be squared). However, in the summation on the left above, at most $a$ of the above terms are zero. This is true as $u_{i+1}^{(k)} - u_i^{(k)} = X_{i+1} w_{i+1}^{(k)} - X_{i+1} w_{i+1}^{(k-1)}, i = 1, \ldots, p-1$, and for large enough values of $k$, as considered by these authors, we will have $w_i^{(k)} = 0$ for all $i \notin A$, as shown in Lemma 1 by Iusem and De Pierro (1990). Thus the last display can be sharpened to

$$\left(\sum_{i=1}^{p-1} \|u_{i+1}^{(k)} - u_i^{(k)}\|_2\right)^2 \leq a \sum_{i=1}^{p-1} \|u_{i+1}^{(k)} - u_i^{(k)}\|_2^2,$$

which allows to define $\sigma = \mu^2/a$. Retracing through the steps above to upper bound $(1/1+\sigma)^{1/2}$, and applying (A.3), then leads to the result as stated in the theorem.

## A.4  Proof of Theorem 3

As in the proof of Theorem 2, we observe that the relationship (A.3) between the Dykstra iterates and coordinate descent iterates allows us to turn a statement about the convergence of the latter into one about convergence of the former. We consider Theorem 3.8 in Deutsch and Hundal (1994), on the asymptotically linear convergence rate of Dykstra's (Hildreth's) algorithm for projecting onto an intersection of halfspaces (we note here, as explained in the proof of Theorem 2, that coordinate descent for the lasso can be equated to Dykstra's algorithm on halfspaces, even though in the original dual formulation, Dykstra's algorithm operates on slabs).

Though the error constant is not explicitly written in the statement of Theorem 3.8 in Deutsch and Hundal (1994)[2], the proofs of Lemma 3.7 and Theorem 3.8 from these authors reveals the following. Define $A = \{i \in \{1, \ldots, p\} : |X_i^T \hat{u}| = \lambda\}$, and enumerate $A = \{i_1, \ldots, i_a\}$ with $i_1 < \ldots < i_a$. As in the proof of Theorem 2, we note that the general position assumption on $X$ allows us to write (almost everywhere in $y \in \mathbb{R}^n$) $A = \text{supp}(\hat{w})$, for the unique lasso solution $\hat{w}$. Also define

$$H_{i_j} = \{v \in \mathbb{R}^n : X_{i_j}^T v = 0\}, \quad \text{for } j = 1, \ldots, a.$$

Deutsch and Hundal (1994) show that, for large enough $k$,

$$\frac{\|u_p^{(k+1)} - \hat{u}\|_2}{\|u_p^{(k)} - \hat{u}\|_2} \leq \max_{\substack{B \subseteq A, \\ B = \{\ell_1, \ldots, \ell_b\}, \\ \ell_1 < \ldots < \ell_b}} \left(1 - \prod_{j=1}^{b-1} \left(1 - c^2\left(H_{\ell_j}, H_{\ell_{j+1}} \cap \cdots \cap H_{\ell_b}\right)\right)\right), \quad (A.4)$$

where $c(L, M)$ denotes the cosine of the angle between linear subspaces $L, M$. Now, to simplify the bound on the right-hand side above, we make two observations. First, we observe that in general

$c(L, M) = c(L^\perp, M^\perp)$ (as in, e.g., Theorem 3.5 of Deutsch and Hundal 1994), so we have

$$
\begin{aligned}
c\Big(H_{\ell_j}, H_{\ell_{j+1}} \cap \cdots \cap H_{\ell_b}\Big) &= c\Big(H_{\ell_j}^\perp, (H_{\ell_{j+1}} \cap \cdots \cap H_{\ell_b})^\perp\Big) \\
&= c\Big(\mathrm{col}(X_{\ell_j}), \mathrm{col}(X_{\{\ell_{j+1},\ldots,\ell_b\}})\Big) \\
&= \frac{\|P_{\{\ell_{j+1},\ldots,\ell_b\}} X_{\ell_j}\|_2}{\|X_{\ell_j}\|_2},
\end{aligned}
$$

where in the last line we used that the cosine of the angle between subspaces has an explicit form, when one of these subspaces is 1-dimensional. Second, we observe that the maximum in (A.4) is actually achieved at $B = A$, since the cosine of the angle between a 1-dimensional subspace and a second subspace can only increase when the second subspace is made larger. Putting these two facts together, and using (A.3), establishes the result in the theorem.

## A.5 Derivation details for (13), (14) and proof of Theorem 4

By rescaling, problem (12) can be written as

$$
\min_{\tilde{u} \in \mathbb{R}^{nd}} \|\tilde{y} - \tilde{u}\|_2^2 \quad \text{subject to} \quad \tilde{u} \in \tilde{C}_0 \cap (\tilde{C}_1 \times \cdots \times \tilde{C}_d), \tag{A.5}
$$

where $\tilde{y} = (\sqrt{\gamma_1} y, \ldots, \sqrt{\gamma_d} y) \in \mathbb{R}^{nd}$, and

$$
\tilde{C}_0 = \{(v_1, \ldots, v_d) \in \mathbb{R}^{nd} : v_1/\sqrt{\gamma_1} = \cdots = v_d/\sqrt{\gamma_d}\} \quad \text{and} \quad \tilde{C}_i = \sqrt{\gamma_i} C_i, \quad \text{for } i = 1, \ldots, d.
$$

The iterations in (13) then follow by applying Dykstra's algorithm to (A.5), transforming the iterates back to the original scale (so that the projections are all in terms of $C_0, C_1, \ldots, C_d$), and recognizing that the sequence say $z_0^{(k)}$, $k = 1, 2, 3, \ldots$ that would usually accompany $u_0^{(k)}$, $k = 1, 2, 3, \ldots$ is not needed because $C_0$ is a linear subspace.

As for the representation (14), it can be verified via a simple inductive argument that the Dykstra iterates in (13) satisfy, for all $k = 1, 2, 3, \ldots$,

$$
u_0^{(k)} = y - \sum_{i=1}^{d} \gamma_i z_i^{(k-1)}, \quad i = 1, \ldots, d.
$$

Also, as shown in the proof of Theorem 6 in Section A.9 below, the image of the residual projection operator $\mathrm{Id} - P_{C_i}$ is contained in the column span of $X_i$, for each $i = 1, \ldots, d$. This means that we can parametrize the Dykstra iterates, for $k = 1, 2, 3, \ldots$, as

$$
u_0^{(k)} = y - \sum_{i=1}^{d} \gamma_i X_i \tilde{w}_i^{(k-1)} \quad \text{and} \quad z_i^{(k)} = X_i \tilde{w}_i^{(k)}, \quad i = 1, \ldots, d,
$$

for some sequence $\tilde{w}_i^{(k)}$, $i = 1, \ldots, d$, and $k = 1, 2, 3, \ldots$. The $z$-updates in (13) then become

$$
X_i \tilde{w}_i^{(k)} = (\mathrm{Id} - P_{C_i})(u_0^{(k)} + X_i \tilde{w}_i^{(k-1)}), \quad i = 1, \ldots, d,
$$

and by Lemma 1, this is equivalent to

$$
\tilde{w}_i^{(k)} = \operatorname*{argmin}_{\tilde{w}_i \in \mathbb{R}^{p_i}} \frac{1}{2}\|u_0^{(k)} + X_i \tilde{w}_i^{(k-1)} - X_i \tilde{w}_i\|_2^2 + h_i(\tilde{w}_i), \quad i = 1, \ldots, d.
$$

Rescaling once more, to $w_i^{(k)} = \gamma_i \tilde{w}_i^{(k)}$, $i = 1, \ldots, d$ and $k = 1, 2, 3, \ldots$, gives the iterations (14).

Lastly, we give a proof of Theorem 4. We can write the second set in the 2-set best approximation problem (A.5) as

$$
\tilde{C}_1 \times \cdots \times \tilde{C}_d = (M^T)^{-1}\Big(\tilde{D}_1 \times \cdots \times \tilde{D}_d\Big),
$$

where $\tilde{D}_i = \sqrt{\gamma_i} D_i$, $i = 1, \ldots, d$, and

$$
M = \begin{pmatrix} X_1 & 0 & \ldots & 0 \\ 0 & X_2 & \ldots & 0 \\ \vdots & & & \\ 0 & 0 & \ldots & X_d \end{pmatrix} \in \mathbb{R}^{nd \times p}.
$$

The duality result established in Theorem 1 can now be applied directly to (A.5). (We note that the conditions of the theorem are met because the matrix $M$, as defined above, has full column rank as each $X_i$, $i = 1, \ldots, d$ does.) Writing $h_S(v) = \max_{s \in S} \langle s, v \rangle$ for the support function of a set $S$, the theorem tells us that the dual of (A.5) is

$$\min_{\tilde{w} \in \mathbb{R}^p, \, \tilde{\alpha} \in \mathbb{R}^{nd}} \frac{1}{2} \|\tilde{y} - M\tilde{w} - \tilde{\alpha}\|_2^2 + h_{\tilde{D}_1 \times \cdots \times \tilde{D}_d}(\tilde{w}) + h_{\tilde{C}_0}(\tilde{\alpha}), \tag{A.6}$$

and the solutions in (A.5) and (A.6), denoted by $\tilde{u}^*$ and $\tilde{w}^*, \tilde{\alpha}^*$ respectively, are related by

$$\tilde{u}_i^* = \sqrt{\gamma_i} y - X_i \tilde{w}_i^* - \tilde{\alpha}_i^*, \quad i = 1, \ldots, d. \tag{A.7}$$

Rescaling to $(\bar{w}_1, \ldots, \bar{w}_d) = (\tilde{w}_1/\sqrt{\gamma}_1, \ldots, \tilde{w}_d/\sqrt{\gamma}_d)$ and $(\bar{\alpha}_1, \ldots, \bar{\alpha}_d) = (\tilde{\alpha}_1/\sqrt{\gamma}_1, \ldots, \tilde{\alpha}_d/\sqrt{\gamma}_d)$, the problem (A.6) becomes

$$\min_{\bar{w} \in \mathbb{R}^p, \, \bar{\alpha} \in \mathbb{R}^{nd}} \frac{1}{2} \sum_{i=1}^d \gamma_i \|y - X_i \bar{w}_i - \bar{\alpha}_i\|_2^2 + \sum_{i=1}^d h_i(\gamma_i \bar{w}_i) + h_{C_0}(\gamma_1 \bar{\alpha}_1, \ldots, \gamma_d \bar{\alpha}_d)$$

$$\iff \min_{\bar{w} \in \mathbb{R}^p, \, \bar{\alpha} \in \mathbb{R}^{nd}} \frac{1}{2} \sum_{i=1}^d \gamma_i \|y - X_i \bar{w}_i - \bar{\alpha}_i\|_2^2 + \sum_{i=1}^d h_i(\gamma_i \bar{w}_i) \text{ subject to } \sum_{i=1}^d \gamma_i \bar{\alpha}_i = 0$$

$$\iff \min_{\bar{w} \in \mathbb{R}^p} \frac{1}{2} \left\| y - \sum_{i=1}^d \gamma_i X_i \bar{w}_i \right\|_2^2 + \sum_{i=1}^d h_i(\gamma_i \bar{w}_i).$$

In the second line we rewrote the support function of $D_1 \times \cdots \times D_d$ as a sum and that of $C_0$ as a constraint; in the third line we optimized over $\bar{\alpha}$ and used $\sum_{i=1}^d \gamma_i = 1$. Clearly, the problem in the last display is exactly the regularized regression problem (2) after another rescaling, $(w_1, \ldots, w_d) = (\gamma_1 \bar{w}_1, \ldots, \gamma_d \bar{w}_d)$. That the solutions in (A.5), (A.6) are related by (A.7) implies that the solutions $\hat{u}, \hat{w}$ in (12), (2) are related by

$$\hat{u}_1 = \cdots = \hat{u}_d = y - X\hat{w}.$$

By Lemma 4.9 in Han (1988), we know that when (A.6) has a unique solution, the dual iterates in Dykstra's algorithm for (A.5) converge to the solution in (A.6). Equivalently, when (2) has a unique solution, the iterates $w^{(k)}$, $k = 1, 2, 3, \ldots$ in (14) converge to the solution in (2).

Also, by Theorem 4.7 in Han (1988), if

$$\text{int} \bigcap_{i=1}^d (X_i^T)^{-1}(D_i) \neq \emptyset,$$

then the sequence $w^{(k)}$, $k = 1, 2, 3, \ldots$ produced by (14) has at least one accumulation point, and each accumulation point solves (2). Moreover, the sequence $Xw^{(k)}$, $k = 1, 2, 3, \ldots$ converges to $X\hat{w}$, the unique fitted value at optimality in (2). In fact, Theorem 4.8 in Han (1988) shows that a weaker condition can be used when some of the sets are polyhedral. In particular, if $D_1, \ldots, D_q$ are polyhedral, then the condition in the above display can be weakened to

$$(X_1^T)^{-1}(D_1) \cap \cdots \cap (X_q^T)^{-1}(D_q) \cap \text{int}(X_{q+1}^T)^{-1}(D_{q+1}) \cap \cdots \cap \text{int}(X_d^T)^{-1}(D_d) \neq \emptyset,$$

and the same conclusion applies.

## A.6 Asymptotic linear convergence of the parallel-Dykstra-CD iterations for the lasso problem

Here we state and prove a result on the convergence rate of the parallel-Dykstra-CD iterations (14) for the lasso problem (9).

**Theorem A.1** (**Adaptation of Iusem and De Pierro 1990**). *Assume the columns of $X \in \mathbb{R}^{n \times p}$ are in general position, and $\lambda > 0$. Then parallel-Dykstra-CD (14) for the lasso (9) has an asymptotically linear convergence rate, in that for large enough $k$, using the notation of Theorem 2,*

$$\frac{\|w^{(k+1)} - \hat{w}\|_\Sigma}{\|w^{(k)} - \hat{w}\|_\Sigma} \leq \left( \frac{2a/\gamma_{\min}}{(2a/\gamma_{\min} + \lambda_{\min}(X_A^T X_A)/\max_{i \in A} \|X_i\|_2^2} \right)^{1/2}, \tag{A.8}$$

*where $\gamma_{\min} = \min_{i=1,\ldots,p} \gamma_i \leq 1/p$ is the minimum of the weights.*

We note that the parallel bound (A.8) is worse than the serial bound (10), because the former relies on a quantity $2a/\gamma_{\min} \geq 2pa$ where the latter relies on $a^2$. We conjecture that the bound (A.8) can be sharpened, by modifying the parallel algorithm so that we renormalize the weights in each cycle after excluding the weights from zero coefficients.

*Proof.* The proof is similar to that for Theorem 2, given in Section A.3. By Theorem 2 in Iusem and De Pierro (1990), for large enough $k$, the iterates of (13) satisfy

$$\frac{\|u_0^{(k+1)} - \hat{u}\|_2}{\|u_0^{(k)} - \hat{u}\|_2} \leq \left(\frac{1}{1+\sigma}\right)^{1/2},$$

where $\sigma = \mu^2/((1/\gamma_{\min} - 1)(2 - \gamma_{\min})) \geq \mu^2 \gamma_{\min}/2$, and $\mu > 0$ is exactly as in Section A.3. The derivation details for (13), (14) in the last section revealed that the iterates from these two algorithms satisfy

$$z_i^{(k)} = X_i w_i^{(k)}/\gamma_i, \quad i = 1, \ldots, d \quad \text{and} \quad u_0^{(k+1)} = y - Xw^{(k)}, \quad \text{for } k = 1, 2, 3, \ldots,$$

hence

$$\|u_0^{(k+1)} - \hat{u}\|_2 = \|Xw^{(k)} - X\hat{w}\|_2 = \|w^{(k)} - \hat{w}\|_\Sigma, \quad \text{for } k = 1, 2, 3, \ldots,$$

which gives the result. $\square$

## A.7 Derivation details for (15) and proof of Theorem 5

Recall that (12) can be rewritten as in (A.5). The latter is a 2-set best approximation problem, and so an ADMM algorithm takes the form of (7) in Section 2. Applying this to (A.5), and transforming the iterates back to their original scale, we arrive at the following ADMM algorithm. We initialize $u_1^{(0)} = \cdots = u_d^{(0)} = 0$, $z_1^{(0)} = \cdots = z_d^{(0)} = 0$, and repeat for $k = 1, 2, 3, \ldots$:

$$
\begin{aligned}
u_0^{(k)} &= \frac{y}{1+\rho} + \frac{\rho}{1+\rho} \sum_{i=1}^{d} \gamma_i(u_i^{(k-1)} - z_i^{(k-1)}), \\
u_i^{(k)} &= P_{C_i}(u_0^{(k)} + z_i^{(k-1)}), \\
z_i^{(k)} &= z_i^{(k-1)} + u_0^{(k)} - u_i^{(k)},
\end{aligned}
\left.\begin{aligned} \\ \\ \end{aligned}\right\} \quad \text{for } i = 1, \ldots, d.
\tag{A.9}
$$

Basically the same arguments as those given in Section A.5, where we argued that (13) is equivalent to (14), now show that (A.9) is equivalent to (15). Note that in the latter algorithm, we have slightly rewritten the algorithm parameters, by using the notation $\rho_i = \rho \gamma_i$, $i = 1, \ldots, d$. That the parallel-ADMM-CD iterations (15) are equivalent to the parallel-Dykstra-CD iterations (14) follows from the equivalence of the 2-set Dykstra iterations (13) and ADMM iterations (A.9), which, recalling the discussion in Section 2, follows from the fact that $\tilde{C}_0$ is a linear subspace and $\tilde{y} \in \tilde{C}_0$ (i.e., $C_0$ is a linear subspace and $(y, \ldots, y) \in C_0$).

The proof of Theorem 5 essentially just uses the duality established in the proof of Theorem 4 in Section A.5, and invokes standard theory for ADMM from Gabay (1983); Eckstein and Bertsekas (1992); Boyd et al. (2011). As shown previously, the dual of (A.5) is (A.6), and by, e.g., the result in Section 3.2 of Boyd et al. (2011), which applies because

$$\|\tilde{y} - \tilde{u}\|_2^2, I_{\tilde{C}_0}(\tilde{u}), I_{\tilde{C}_1 \times \cdots \times \tilde{C}_d}(\tilde{u})$$

are closed, convex functions of $\tilde{u}$, the scaled dual iterates in the ADMM algorithm for (A.5) converge to a solution in (A.6), or equivalently, the iterates $\rho_i z_i^{(k)} = X_i w_i^{(k)}$, $i = 1, \ldots, d$, $k = 1, 2, 3, \ldots$ in (A.9) converge to the optimal fitted values $X_i \hat{w}_i$, $i = 1, \ldots, d$ in (2), or equivalently, the sequence $w^{(k)}$, $k = 1, 2, 3, \ldots$ in (15) converges to a solution in (2).

## A.8 Details of the experimental setup in Figure 1

Figure 1 displays results from numerical simulations comparing serial parallel coordinate descent (5) to parallel-Dykstra-CD (14) and parallel-ADMM-CD (15) for the lasso problem. Our simulation setup was simple, and the goal was to investigate the basic behavior of the new parallel proposals,

and not to investigate performance at large-scale nor compare to state-of-the art implementations of coordinate descent for the lasso (ours was a standard implementation with no speedup tricks—like warm starts, screening rules, or active set optimization—employed).

We considered a regression setting with $n = 200$ observations and $p = 500$ predictors. Denoting by $x_i \in \mathbb{R}^p$ denotes the $i$th row of $X^{n \times p}$, the data was drawn according to the Gaussian linear model

$$x_i \sim N(0, I_{p \times p}) \quad \text{and} \quad y_i \sim N(x_i^T \beta_0, 1) \quad \text{i.i.d., for } i = 1, \ldots, n,$$

where $\beta_0 \in \mathbb{R}^p$ had its first 20 components equal to 1, and the rest 0. We computed solutions to the lasso problem (9) at $\lambda = 5$, over 30 draws of data $X, y$ from the above model. At this value of $\lambda$, the lasso solution $\hat{w}$ had an average of 151.4 nonzero components over the 30 trials. (Larger values of $\lambda$ resulted in faster convergence for all algorithms and we found the comparisons more interesting at this smaller, more challenging value of $\lambda$.)

The figure shows the suboptimality, i.e., achieved criterion value minus optimal criterion value, as a function of iteration number, for:

- the usual serial coordinate descent iterations (5), in black;
- the parallel-Dykstra-CD iterations (14) with $\gamma_1 = \cdots = \gamma_p = 1/p$, in red;
- the parallel-ADMM-CD iterations (15) with $\rho_1 = \cdots = \rho_p = 1/p$ and 3 different settings of $\rho = \sum_{i=1}^p \rho_i$, namely $\rho = 10, 50, 200$, in green, blue, and purple respectively.

(Recall that for $\rho = 1$, parallel-ADMM-CD and parallel-Dykstra-CD are equivalent.) Thin colored lines in the figures denote the suboptimality curves for individual lasso problem instances, and thick colored lines represent the average suboptimality curves over the 30 total instances. In all instances, suboptimality is measured with respect to the criterion value achieved by the least angle regression algorithm (Efron et al., 2004), which is a direct algorithm for the lasso and should return the exact solution up to computer precision.

The left panel of the figure displays the suboptimality curves as a function of raw iteration number, which for the parallel methods (14), (15) would correspond to running these algorithms in a naive serial mode. In the right panel, iterations of the parallel methods are counted under a hypothetical "10% efficient" parallel implementation, where $0.1p$ updates of the $p$ total updates in (14), (15) are able to be computed at the cost of 1 serial update in (5). (A "100% efficient" implementation would mean that all $p$ updates in (14), (15) could be performed at the cost of 1 serial update in (5), which, depending on the situation, may certainly be unrealistic, due to a lack of available parallel processors, synchronization issues, etc.) While the parallel methods display much worse convergence based on raw iteration number, they do offer clear benefits in the 10% parallelized scenario. Also, it seems that a larger value of $\rho$ generally leads to faster convergence, though the benefits of taking $\rho = 200$ over $\rho = 50$ are not quite as clear (and for values of $\rho$ much larger than 200, performance degrades).

## A.9 Proof of Theorem 6

First, we establish the following generalization of Lemma 1.

**Lemma A.2.** *Let $f$ be a closed, strictly convex, differentiable function. Assume that $X_i \in \mathbb{R}^{n \times p_i}$ has full column rank, and let $h_i(v) = \max_{d \in D_i} \langle d, v \rangle$ for a closed, convex set $D_i \subseteq \mathbb{R}^{p_i}$. Then for $C_i = (X_i^T)^{-1}(D_i) \subseteq \mathbb{R}^n$, and any $a \in \mathbb{R}^n$,*

$$\hat{w}_i = \operatorname*{argmin}_{w_i \in \mathbb{R}^{p_i}} f(a + X_i w_i) + h_i(w_i) \iff X_i \hat{w}_i = \left( \nabla g - \nabla g \circ P_{C_i}^g \right) \left( \nabla g^*(-a) \right),$$

*where $g(v) = f^*(-v)$.*

*Proof.* We begin by analyzing the optimality condition that characterizes the Bregman projection $\hat{u}_i = P_{C_i}^g(x) = \operatorname{argmin}_{c \in C_i} g(c) - g(x) - \langle \nabla g(x), c - x \rangle$, namely

$$\nabla g(x) - \nabla g(\hat{u}_i) \in \partial I_{C_i}(\hat{u}_i).$$

Defining $\hat{z}_i = \nabla g(x) - \nabla g(\hat{u}_i) = (\nabla g - \nabla g \circ P_{C_i}^g)(x)$, this becomes

$$\hat{z}_i \in \partial I_{C_i} \left( \nabla g^* \left( \nabla g(x) - \hat{z}_i \right) \right),$$

where we have used the fact that $\nabla g^* = (\nabla g)^{-1}$, allowing us to rewrite the relationship between $\hat{u}_i, \hat{z}_i$ as $\hat{u}_i = \nabla g^*(\nabla g(x) - \hat{z}_i)$. And lastly, substituting $g(v) = f^*(-v)$ (and $g^*(v) = f(-v)$) the optimality condition reads

$$\hat{z}_i \in \partial I_{C_i}\Big(-\nabla f\big(\nabla f^*(-x) + \hat{z}_i\big)\Big). \tag{A.10}$$

Now we investigate the claim in the lemma. By subgradient optimality,

$$\hat{w}_i = \underset{w_i \in \mathbb{R}^{p_i}}{\text{argmin}}\ f(a + X_i w_i) + h_i(w_i) \iff -X_i^T(\nabla f)(a + X_i \hat{w}_i) \in \partial h_i(\hat{w}_i)$$

$$\iff \hat{w}_i \in \partial h_i^*\Big(-X_i^T(\nabla f)(a + X_i \hat{w}_i)\Big)$$

$$\iff X_i \hat{w}_i \in X_i \partial h_i^*\Big(-X_i^T(\nabla f)(a + X_i \hat{w}_i)\Big).$$

The second line follows from the fact that, for a closed, convex function $g$, subgradients of $g$ and of $g^*$ are related via $x \in \partial f(y) \iff y \in \partial g^*(x)$; the third line follows from the fact that $X_i$ has full column rank. Note that $h_i^* = I_{D_i}$, the indicator function of $D_i$, and denote $h_{C_i}(v) = \sup_{c \in C_i}\langle c, v\rangle$. Then following from the last display, by the chain rule,

$$\hat{w}_i = \underset{w_i \in \mathbb{R}^{p_i}}{\text{argmin}}\ f(a + X_i w_i) + h_i(w_i) \iff X_i \hat{w}_i \in \partial h_{C_i}^*\Big(-\nabla f(a + X_i \hat{w}_i)\Big),$$

because $h_{C_i}^* = I_{C_i} = I_{D_i} \circ X_i^T$. Applying the previous fact in (A.10) on Bregman projections gives

$$\hat{w}_i = \underset{w_i \in \mathbb{R}^{p_i}}{\text{argmin}}\ f(a + X_i w_i) + h_i(w_i) \iff X_i \hat{w}_i = \Big(\nabla g - \nabla g \circ P_{C_i}^g\Big)(x)$$

for $a = \nabla f^*(-x) = -\nabla g(x)$, i.e., for $x = \nabla g^*(-a)$. This completes the proof of the lemma. □

We are ready for the proof of the theorem. We start with the claim about duality between (16), (18). Standard arguments in convex analysis show that the Lagrange dual of (16) is

$$\max_{u \in \mathbb{R}^n}\ -f^*(-u) - \sum_{i=1}^d h_i^*(X_i^T u),$$

where $f^*$ is the conjugate of $f$ and $h_i^* = I_{D_i}$ the conjugate of $h_i$, $i = 1, \ldots, d$, with the relationship between the primal $\hat{w}$ and dual $\hat{u}$ solutions being $\hat{u} = -\nabla f(X\hat{w})$. Written in equivalent form, the dual problem is

$$\min_{u \in \mathbb{R}^n}\ f^*(-u) \quad \text{subject to} \quad u \in C_1 \cap \cdots \cap C_d.$$

Recalling $g(v) = f^*(-v)$, and $b = -\nabla f(0)$, we have by construction

$$D_g(u, b) = g(u) - g(b) - \langle \nabla g(b), u - b\rangle = f^*(-u) - f^*(\nabla f(0)),$$

where we have used the fact that $\nabla g(b) = -\nabla f^*(\nabla f(0)) = 0$, as $\nabla f^* = (\nabla f)^{-1}$. Therefore the above dual problem, in the second to last display, is equivalent to (18), establishing the claim.

Now we proceed to the claim about the equivalence between Dykstra's algorithm (19) and coordinate descent (17). We note that a simple inductive argument shows that the Dykstra iterates satisfy, for all $k = 1, 2, 3, \ldots$,

$$\nabla g(u_i^{(k)}) = -\sum_{j \le i} z_j^{(k)} - \sum_{j > i} z_j^{(k-1)}, \quad i = 1, \ldots, d.$$

We also note that, for $i = 1, \ldots, d$, the image of $\nabla g - \nabla g \circ P_{C_i}^g$ is contained in the column span of $X_i$. To see this, write $\hat{u}_i = P_{C_i}^g(a)$, and recall the optimality condition for the Bregman projection,

$$\langle \nabla g(\hat{u}_i) - \nabla g(a), c - \hat{u}_i\rangle \ge 0, \quad c \in C_i.$$

If $\langle \nabla g(\hat{u}_i) - \nabla g(a), \delta\rangle \ne 0$ for some $\delta \in \text{null}(X_i^T)$, supposing without a loss of generality that this inner product is negative, then the above optimality condition breaks for $c = \hat{u}_i + \delta \in C_i$. Thus we have shown by contradiction that $\nabla g(a) - \nabla g(\hat{u}_i) \perp \text{null}(X_i^T)$, i.e., $\nabla g(a) - \nabla g(\hat{u}_i) \in \text{col}(X_i)$, the desired fact.

Putting together the last two facts, we can write the Dykstra iterates, for $k = 1, 2, 3, \ldots$, as

$$z_i^{(k)} = X_i \tilde{w}_i^{(k)} \quad \text{and} \quad \nabla g(u_i^{(k)}) = -\sum_{j \leq i} X_j \tilde{w}_j^{(k)} - \sum_{j > i} X_j \tilde{w}_j^{(k-1)}, \quad \text{for } i = 1, \ldots, d,$$

for some sequence $\tilde{w}_i^{(k)}$, $i = 1, \ldots, d$, and $k = 1, 2, 3, \ldots$. In this parametrization, the $z$-updates in the Dykstra iterations (19) are thus

$$X_i \tilde{w}_i^{(k)} = \left( \nabla g - \nabla g \circ P_{C_i}^g \right) \left( \nabla g^* \left( -\sum_{j < i} X_j \tilde{w}_j^{(k)} - \sum_{j > i} X_j \tilde{w}_j^{(k-1)} \right) \right), \quad i = 1, \ldots, d,$$

where we have used the fact that $\nabla g^* = (\nabla g)^{-1}$. Invoking Lemma A.2, we know that the above is equivalent to

$$\tilde{w}_i^{(k)} = \operatorname*{argmin}_{\tilde{w}_i \in \mathbb{R}^{p_i}} f \left( \sum_{j < i} X_j \tilde{w}_j^{(k)} + \sum_{j > i} X_j \tilde{w}_j^{(k-1)} + X_i \tilde{w}_i \right) + h_i(\tilde{w}_i), \quad i = 1, \ldots, d,$$

which are exactly the coordinate descent iterations (5). It is easy to check that the initial conditions for the two algorithms also match, and hence $\tilde{w}_i^{(k)} = w_i^{(k)}$, for all $i = 1, \ldots, d$ and $k = 1, 2, 3, \ldots$, completing the proof.

## A.10 Derivation details for (20), (21)

We first consider parallelization of projection algorithms for the best Bregman-approximation problem (18). For simplicity and without a loss of generality we will assume an equal weighting $\gamma_1 = \cdots = \gamma_d = 1/d$ throughout; the arguments for arbitrary weights are similar. As in the Euclidean projection case, to derive parallel algorithms for (18) we will turn to a product space reparametrization, namely,

$$\min_{u \in \mathbb{R}^{nd}} D_{\tilde{g}}(u, \tilde{b}) \quad \text{subject to} \quad u \in C_0 \cap (C_1 \times \cdots \times C_d), \tag{A.11}$$

where $C_0 = \{(u_1, \ldots, u_d) \in \mathbb{R}^{nd} : u_1 = \cdots = u_d\}$, $\tilde{b} = (b, \ldots, b) \in \mathbb{R}^{nd}$, and we define the function $\tilde{g} : \mathbb{R}^{nd} \to \mathbb{R}$ by $\tilde{g}(u_1, \ldots, u_d) = \sum_{i=1}^{d} g(u_i)$.

Dykstra's algorithm for the 2-set problem (A.11) sets $u_1^{(0)} = \cdots = u_d^{(0)} = b$, $r_1^{(0)} = \cdots = r_d^{(0)} = 0$, and $z_1^{(0)} = \cdots = z_d^{(0)} = 0$, then repeats for $k = 1, 2, 3, \ldots$:

$$
\begin{aligned}
u_0^{(k)} &= \operatorname*{argmin}_{u_0 \in \mathbb{R}^n} g(u_0) - \frac{1}{d} \sum_{i=1}^{d} \left\langle \nabla g(u_i^{(k-1)}) + r_i^{(k-1)}, u_0 \right\rangle, \\
\left.
\begin{aligned}
r_i^{(k)} &= \nabla g(u_i^{(k-1)}) + r_i^{(k-1)} - \nabla g(u_0^{(k)}), \\
u_i^{(k)} &= (P_{C_i}^g \circ \nabla g^*) \left( \nabla g(u_0^{(k)}) + z_i^{(k-1)} \right), \\
z_i^{(k)} &= \nabla g(u_0^{(k)}) + z_i^{(k-1)} - \nabla g(u_i^{(k)}),
\end{aligned}
\right\} &\quad \text{for } i = 1, \ldots, d.
\end{aligned}
\tag{A.12}
$$

Now we will rewrite the above iterations, under $b = -\nabla f(0)$, where $g(v) = f^*(-v)$. The $u_0$-update in (A.12) is defined by the Bregman projection of

$$\left( \nabla g^* \left( \nabla g(u_1^{(k-1)}) + r_1^{(k-1)} \right), \ldots, \nabla g^* \left( \nabla g(u_d^{(k-1)}) + r_d^{(k-1)} \right) \right) \in \mathbb{R}^{nd}$$

onto the set $C_0$, with respect to the function $\tilde{g}$. By first-order optimality, this update can be rewritten as

$$\nabla g(u_0^{(k)}) = \frac{1}{d} \sum_{i=1}^{d} \left( \nabla g(u_i^{(k-1)}) + r_i^{(k-1)} \right).$$

Plugging in the form of the $r$-updates, and using a simple induction, the above can be rewritten as

$$\nabla g(u_0^{(k)}) = \frac{1}{d} \sum_{i=1}^{d} \sum_{\ell=1}^{k-1} \left( \nabla g(u_i^{(\ell)}) - \nabla g(u_0^{(\ell)}) \right)$$

$$= \nabla g(b) + \frac{1}{d} \sum_{i=1}^{d} \sum_{\ell=1}^{k-1} (z_i^{(\ell-1)} - z_i^{(\ell)})$$

$$= -\frac{1}{d} \sum_{i=1}^{d} z_i^{(k)},$$

where in the second line we used the relationship given by $z$-updates and recalled the initializations $u_1^{(0)} = \cdots = u_d^{(0)} = b$, and in the third line we used $\nabla g(b) = \nabla g(-\nabla f(0)) = \nabla g(\nabla g^*(-0)) = 0$. Similar arguments as those given in the proof of Theorem 6 in Section A.9 show that the $u$-updates and $z$-updates in (A.12) can be themselves condensed to

$$X_i \tilde{w}_i^{(k)} = \left( \nabla g - \nabla g \circ P_{C_i}^g \right) \left( \nabla g^* \left( \nabla g(u_0^{(k)}) + X_i \tilde{w}_i^{(k)} \right) \right), \quad i = 1, \ldots, d,$$

for a sequence $\tilde{w}_i^{(k)}$, $i = 1, \ldots, d$, and $k = 1, 2, 3, \ldots$, related by $z_i^{(k)} = X_i \tilde{w}_i^{(k)}$, $i = 1, \ldots, d$, and $k = 1, 2, 3, \ldots$. By Lemma A.2, the above is equivalent to

$$\tilde{w}_i^{(k)} = \operatorname*{argmin}_{\tilde{w}_i \in \mathbb{R}^{p_i}} \ f\left( -\nabla g(u_0^{(k)}) - X_i \tilde{w}_i^{(k)} + X_i \tilde{w}_i \right) + h_i(\tilde{w}_i), \quad i = 1, \ldots, d,$$

Rescaling to $w_i^{(k)} = \tilde{w}_i^{(k)}/d$, $i = 1, \ldots, d$, $k = 1, 2, 3, \ldots$, the above displays show that the Dykstra iterations (A.12) can be rewritten quite simply as:

$$w_i^{(k)} = \operatorname*{argmin}_{w_i \in \mathbb{R}^{p_i}} \ f\left( X w^{(k)} - d X_i w_i^{(k)} + d X_i w_i \right) + h_i(d w_i), \quad i = 1, \ldots, d, \qquad \text{(A.13)}$$

for $k = 1, 2, 3, \ldots$, with the initialization being $w^{(0)} = 0$. This is precisely our parallel-Dykstra-CD algorithm (20) for (16), under equal weights $\gamma_1 = \cdots = \gamma_d = 1/d$.

Meanwhile, ADMM for the 2-set problem (A.11) is defined based on the augmented Lagrangian

$$L(u_0, \ldots, u_d, z_1, \ldots, z_d) = d\big( g(u_0) - \langle \nabla g(b), u_0 \rangle \big) +$$

$$\sum_{i=1}^{d} \left( I_{C_i}(u_i) + \rho \|u_0 - u_i + z_i\|_2^2 \right) - \rho \sum_{i=1}^{d} \|z_i\|_2^2.$$

Initializing $u_1^{(0)} = \cdots = u_d^{(0)} = 0$, $z_1^{(0)} = \cdots = z_d^{(0)} = 0$, we repeat for $k = 1, 2, 3, \ldots$:

$$u_0^{(k)} = \operatorname*{argmin}_{u_0 \in \mathbb{R}^n} \ g(u_0) - \langle \nabla g(b), u_0 \rangle + \frac{\rho}{d} \sum_{i=1}^{d} \|u_0 - u_i^{(k-1)} + z_i^{(k-1)}\|_2^2,$$

$$\left.\begin{aligned} u_i^{(k)} &= P_{C_i}(u_0^{(k)} + z_i^{(k-1)}), \\ z_i^{(k)} &= z_i^{(k-1)} + u_0^{(k)} - u_i^{(k)}, \end{aligned}\right\} \quad \text{for } i = 1, \ldots, d. \qquad \text{(A.14)}$$

Again using $b = -\nabla f(0)$, with $g(v) = f^*(-v)$, we will now rewrite the above iterations. Precisely as in the connection between (A.9) and (15) in the quadratic case, as discussed in Section A.7, the $u$-updates and $z$-updates here reduce to

$$w_i^{(k)} = \operatorname*{argmin}_{\tilde{w}_i \in \mathbb{R}^{p_i}} \ \frac{1}{2} \left\| u_0^{(k)} + X_i \tilde{w}_i^{(k-1)} - X_i \tilde{w}_i \right\|_2^2 + h_i(\tilde{w}_i) \quad i = 1, \ldots, d,$$

where $\tilde{w}_i^{(k)}$, $i = 1, \ldots, d$, $k = 1, 2, 3, \ldots$ satisfies $z_i^{(k)} = X_i \tilde{w}_i^{(k)}$, $i = 1, \ldots, d$, $k = 1, 2, 3, \ldots$. The $u_0$-update here is characterized by

$$\nabla g(u_0^{(k)}) = \frac{\rho}{d} \sum_{i=1}^{d} (u_i^{(k-1)} - z_i^{(k-1)}) - \rho u_0^{(k)},$$

where we have used $\nabla g(b) = 0$, or equivalently,

$$u_0^{(k)} = -\nabla f\left( \rho u_0^{(k)} - \frac{\rho}{d} \sum_{i=1}^{d} (u_i^{(k-1)} - z_i^{(k-1)}) \right),$$

where we have used $\nabla g^* = (\nabla g)^{-1}$ and $g^*(v) = f(-v)$, and lastly

$$u_0^{(k)} = -\nabla f\left( \rho(u_0^{(k)} - u_0^{(k-1)}) - \frac{\rho}{d} X(\tilde{w}^{(k-2)} - 2\tilde{w}^{(k-1)}) \right),$$

by plugging in the form of the $u$-updates, and recalling $z_i^{(k)} = X_i \tilde{w}_i^{(k)}$, $i = 1, \ldots, d$, $k = 1, 2, 3, \ldots$. Rescaling to $w_i^{(k)} = (\rho/d)\tilde{w}_i^{(k)}$, $i = 1, \ldots, d$, $k = 1, 2, 3, \ldots$, and collecting the last several displays, we have shown that the ADMM iterations (A.14) can be written as:

Find $u_0^{(k)}$ such that:   $u_0^{(k)} = -\nabla f\left( \rho(u_0^{(k)} - u_0^{(k-1)}) - X(\tilde{w}^{(k-2)} - 2\tilde{w}^{(k-1)}) \right),$

$$w_i^{(k)} = \operatorname*{argmin}_{w_i \in \mathbb{R}^{p_i}} \frac{1}{2} \left\| u_0^{(k)} + (d/\rho) X_i w_i^{(k-1)} - (d/\rho) X_i w_i \right\|_2^2 + h_i\big((d/\rho)w_i\big), \quad i = 1, \ldots, d,$$

for $k = 1, 2, 3, \ldots$, with initializations $u_0^{(0)} = 0$, $w^{(-1)} = w^{(0)} = 0$. This is our parallel-ADMM-CD algorithm (21) for (16), when we choose equal augmented Lagrangian parameters $\rho_1 = \cdots = \rho_d = \rho/d$.

## Footnotes

[1] By this we mean that $u_i^{-,(k)} = u_i^{(k)}$ for all $i = 1, \ldots, p$ and $k = 1, 2, 3, \ldots$, if the iterates from Dykstra's algorithm on $C_1^+ \cap C_1^- \cap \ldots \cap C_p^+ \cap C_p^-$ are denoted as $u_i^{+,(k)}, u_i^{-,(k)}, i = 1, \ldots, p$.

[2]The result in Theorem 3.8 of Deutsch and Hundal (1994) is actually written in nonasymptotic form, i.e., it is stated (translated to our notation) that $\|u_d^{(k)} - \hat{u}\|_2 \leq \rho c^k$, for some constants $\rho > 0$ and $0 < c < 1$, and all iterations $k = 1, 2, 3, \ldots$. The error constant $c$ can be explicitly characterized, as we show in the proof of Theorem 3. But the constant $\rho$ cannot be, and in fact, the nonasymptotic error bound in Deutsch and Hundal (1994) is really nothing more than a restatement of the more precise asymptotic error bound developed in the proofs of their Lemma 3.7 and Theorem 3.8. Loosely put, any asymptotic error bound can be transformed into a nonasymptotic one by simply defining a problem-specific constant $\rho$ to be large enough that it makes the bound valid until the asymptotics kick in. This describes the strategy taken in Deutsch and Hundal (1994).