[Reviews · NeurIPS 2017]

Reviewer 1



This paper studies connections/equivalences between Dykstra's algorithm, coordinate descent (à la Gauss-Seidel block alternating minimization). Many of these equivalences were already known (indeed essentially in the optimization literature as stated by the author). The authors investigates this for the soft feasibility (best approximation) problem and block-seperable regularized regression, where the regularizers are positively homogeneous (i.e. supports of closed convex sets containing the origin). The author claims that this is the first work to investigate this for the casen where the design is not unitary. Actually, the extension to arbitrary design is very straightforward (almost trivial) through Fenchel-Rockafellar duality. In fact, positive homogeneity is not even needed as Dykstra's algorithm has been extended to tje proximal setting (beyond indicator functions of closed convex sets). Having said this, I have several concerns on this paper. + My first concern pertains to the actual novelty of the work, which I beleive is quite limited. The manuscript has a flavour of a review paper with some incremental extensions. + The paper contains some important inaccuracies that should be addressed. Other detailed comments are as follows: + (1): intersection of sets should be non-empty. Otherwise many of the statements made do not rigorously hold. + What the author is calling seminorms are actually support functions (symmetry is NOT needed), or equivalently gauges of polar sets. + Extension to the non-euclidean case: several key details are missing for all this to make sense. In particular the notion of a Legendre function. Qualification conditions for strong duality to hold are not stated either.

Reviewer 2



The paper studies the relationship between different splitting algorithms like Dykstra, CD and ADMM. They prove an equivalence between Dykstra and CD that (to the best of my knowledge) is novel in its general form. Clarity: the paper is beautifully written, it is a pleasure to read. Content: While not particularly surprising (this equivalence was already known for a more restricted class of problems, which is acknowledged in the paper), the paper make interesting contributions besides this connection, such as: * Adaptation of Dykstra convergence results (Theorem 2 and 3) to coordinate-descent problems (Lasso). * Parallel CD algorithm. * An extension of this connection to non-quadratic loss functions via Bregman divergences. The paper is correct as far as I can tell, although I only skimmed through the proofs in the appendix. The paper does not contain any experiments, which I didn't see as an issue given that most of the discussed algorithms are well established. However, I would have given a higher rating if it had convincing experiments e.g., on the parallel-Dykstra-CD algorithm. I would be very interested to know from the authors if they are aware of the Dykstra-like proximal splitting method described in [X1] and whether there would be any benefit or any generalization that could come from using this more general algorithm instead of the projection-only Dykstra algorithm that the authors mention. [X1] Combettes, Patrick L., and Jean-Christophe Pesquet. “Proximal splitting methods in signal processing.” https://arxiv.org/pdf/0912.3522.pdf , Algorithm 7.6

Reviewer 3



This is a interesting and well written paper. It provides important insights into coordinate descent and its connections with Dykstra's algorithm. After reading this paper, I agree with the author's sentiment that "the connection between Dykstra's algorithm and coordinate descent is not well-known enough and should be better explored". This paper makes a nice contribution, tying together results sprinkled throughout the optimization literature, and also presents several new additions, putting them together in one cohesive piece of work. I was pleased that the authors included proofs in the supplement, and it was also great that there was a small numerical experiment (again in the supplement).